# Assessing the value of high-resolution rainfall and streamflow data for hydrological modeling: An analysis based on 63 catchments in southeast China

Mahmut Tudaji, Yi Nan*, Fuqiang Tian

5  Department of Hydraulic Engineering & State Key Laboratory of Hydroscience and Engineering, Tsinghua University, Beijing 100084, China

*Correspondence to*: Yi Nan (ny1209@qq.com)

**Abstract.** The temporal resolution of forcing and calibration data substantially influences the performance of hydrological models. This impact varies among regions according to the climatic and landscape characteristics of the watersheds. In this study, we evaluated the benefits of using high-resolution rainfall and streamflow data in hydrological modeling across 63 small-to-medium-scale catchments in Southeast China. We applied rainfall and streamflow data at various resolutions ranging from 1 to 24 hours to drive and calibrate a well-established hydrological model. Our findings reveal that: (1) Utilizing sub-daily rainfall data significantly enhances the accuracy of daily streamflow forecasts, with notable improvements observed when models transition from daily to sub-daily resolutions. (2) Forcing and calibrating the model by rainfall and streamflow data with sub-daily resolution data markedly improve hourly streamflow forecasts compared to daily data, but the enhancements become negligible when the resolution exceeds 6 hours. (3) The advantages of sub-daily resolution data are more pronounced in catchments characterized by smaller drainage areas, significant diurnal streamflow variability, and a greater number of rain gauges. These findings provide basis for a more efficient rainfall and streamflow data acquisition.

## 1 Introduction

Hydrological models are vital for understanding and predicting the dynamics of water resources, as well as occurrences of floods and droughts. The effectiveness of these models heavily depends on the quality and resolution of the data, especially the rainfall used for forcing and measured streamflow for calibration. Traditionally, hydrological modeling has utilized daily or even coarser resolution data, which limits its application for shorter time steps required in scenarios such as flash flood forecasting. To address this limitation, data is often artificially disaggregated from raw time series using mass curves (Blöschl and Sivapalan, 1995) or complex stochastic generators (Creutin and Obled, 1980). However, models based on coarsely resolved or artificially refined data can introduce biases, particularly when forecasting at finer temporal scales, as they may not accurately capture the variability and magnitude of hydrological variables (Younis et al., 2008; Huang et al., 2019).

Rainfall is crucial in driving high-frequency responses in catchments, in contrast to the more gradual changes caused by evapotranspiration (Oudin et al., 2006). The temporal distribution of rainfall profoundly affects runoff patterns, influencing both peak discharge rates (Gabellani et al., 2007) and total runoff volume (Viglione et al., 2010). These effects are primarily due to the nonlinear dynamics of infiltration and runoff generation processes, which typically occur over a timescale of minutes (Blöschl and Sivapalan, 1995; Kandel et al., 2005). Previous studies indicated that models incorporating sub-daily time steps more effectively capture the complexities of infiltration excess and surface runoff, highlighting the importance of peak rainfall rates in accurate rainfall-runoff modeling (Kandel et al., 2004, 2005; Socolofsky et al., 2001; Yu et al., 1998). Additionally, streamflow measurement, a vital component for model calibration, ensures that the quality of calibration data significantly impacts model parameters. Parameters governing slower dynamics exhibit considerable stability across various timescales, whereas those associated with faster dynamics achieve greater accuracy and stability as data resolution improves (Kavetski et al., 2011).

Recent advancements in measurement technologies, including high-frequency automated rain/streamflow gauges and phased array rain-radars, have enabled access to high-resolution rainfall and runoff datasets. Despite these technological advances, the quantitative benefits of high-resolution data in enhancing hydrological model performance remain unclear. For instance, studies on the impact of rainfall data resolution on hydrological models have produced inconsistent results. Research such as Jaehak et al. (2011) suggested that finer temporal resolution significantly improves model simulations, whereas other studies (Kannan et al., 2006; Ficchì et al., 2016) found that greater data resolution does not necessarily lead to better model performance. These variances could be due to factors like the process descriptions in the models, watershed characteristics, and the scale of data aggregation, yet there is a lack of comprehensive research investigating these elements.

This study sought to enhance our understanding of the value of fine time-step hydro-climatic data for hydrological model performance. We designed two experiments focusing on the most common hydrological forecasting timescales—daily and hourly. The impact of high-resolution rainfall and streamflow data was assessed across 63 small-to-medium-scale catchments in Southeastern China, using data resolutions ranging from 1 to 24 hours to drive and calibrate the hydrological models. In addition, we explored factors that influence model performance and evaluated the benefits of high-resolution data from the perspective of watershed characteristics. Specifically, we aim to answer three key questions:

(1) To what extent can sub-daily rainfall data improve daily streamflow simulations?

(2) What is the coarsest resolution of rainfall and streamflow data to provide reliable hourly streamflow simulations?

(3) What factors influence model performance and the value of high-resolution data?

The structure of the remainder of this paper is as follows: Section 2 details the methodology and experimental design, including the selection of the 63 catchments, the hydrological model employed, and the techniques used to quantify the benefits of high-resolution data and to identify influential factors. Section 3 presents the results, offering a comparison of model performance at different temporal resolutions across the selected catchments. We investigate performance variations and discuss possible explanations for these differences. Section 4 examines the implications of measuring rainfall and streamflow and addresses the limitations of this study. Finally, Section 5 provides the concluding remarks.

## 2 Materials and methodology

### 2.1 Catchment set and data

In this study, we utilized a set of 63 small-medium-scale catchments located in Southeastern China (Fig.1). The catchment outlets are geographically dispersed, ranging from 102°E to 119°E longitude and 21°N to 33°N latitude. Predominantly, most of these catchments (57 out of 63) fall within the Yangtze River Basin, while four are situated in the Pearl River Basin, and two in the Southeast Basin. Table 1 presents the statistical summaries of the catchment attributes. The drainage areas of these catchments vary considerably, ranging from 91.5 km² to 5266 km², with an average area of 1528 km². These catchments exhibit significant diversity in climatic conditions and rainfall-runoff relationships, highlighted by a wide range of mean annual rainfall (647 to 2593 mm) and runoff ratios (0.31 to 0.96).

**Table 1. Statistical summaries of the catchment attributes**

| Attributes | Description | Min | Max | Average | Unit |
|---|---|---|---|---|---|
| DRA | Drainage area | 92 | 5266 | 1528 | km² |
| MAR | Mean annual rainfall | 647 | 2593 | 1531 | mm |
| MAQ | Mean annual runoff | 356 | 1571 | 868 | mm |
| QR | Runoff ratio | 0.31 | 0.96 | 0.58 | - |
| RGA | Rainfall gauging area | 27 | 859 | 109 | km² |
| RGN | Number of rainfall gauges | 2 | 64 | 14.87 | - |
| ISTD | Intraday standard deviation | 0.06 | 0.79 | 0.22 | - |
| GOUE | Goodness of uniform estimation | 0.65 | 0.98 | 0.82 | - |

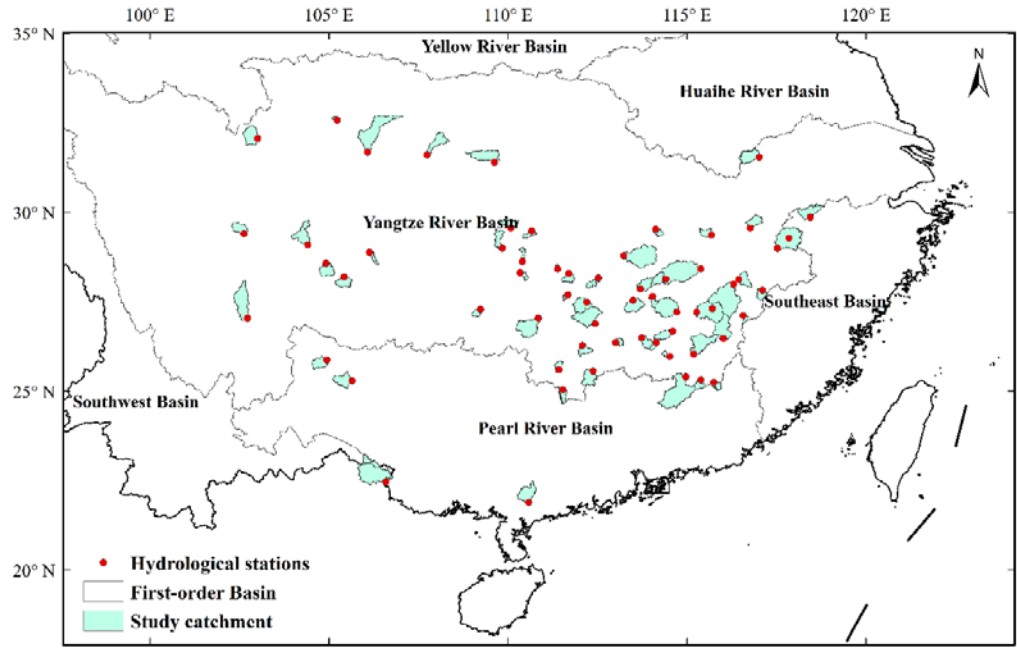

**Figure 1: Geographic distribution of catchments spanning across Southeastern China**

Hydrometeorological data spanning from January 1, 2014, to December 31, 2015, were sourced from the National Rainfall and Hydrological Database, curated by the Information Center of the Ministry of Water Resources (http://xxfb.mwr.cn/sq_dtcx.html). The selection criteria for hydrological stations were established based on several key factors: (1) Catchment size: Focusing on the sub-daily variation of streamflow which is crucial in small and mesoscale catchments, only those with a drainage area of less than 6000 km² were selected. (2) Temporal resolution: The original data varied in resolution from 5 minutes to more than one day. Given the scarcity of stations with complete and continuous time series data, we chose stations with an average resolution—defined by the ratio of the length of the time period to the number of measurements—that exceeded 3650 seconds. This resolution threshold, slightly over one hour, was designed to minimize gaps in hourly time series data. (3) Water level and discharge relationship: The database includes both water level and discharge data. Notably, water level data tends to be completer and more continuous. We utilized the relationship between water level and discharge to infer discharge values for periods only covered by water level data. Stations were chosen based on their provision of discharge data for more than 80% of the time steps that also had water level data and a determination coefficient ($R^2$) of the water level-discharge relationship exceeding 0.95, ensuring the accuracy of these calculations.

The criteria for selecting rainfall data were similar to those for streamflow data. We identified 63 high-quality stations situated within the study catchments from the original database, characterized by an average temporal resolution of slightly over one hour. To generate the areal rainfall data for each catchment, we employed the Thiessen Polygon method (Han and Bray, 2006). The number of rainfall gauges per catchment varied from 2 to 64, averaging 15 stations. Additionally, the rainfall gauging area—calculated as the catchment area divided by the number of stations—ranged from 27 km² to 859 km², with an average of 109 km².

Besides, the DEM in this study was from the MERIT Digital Elevation Model (Yamazaki et al., 2017) with a spatial resolution of 90m. Temperature and potential evapotranspiration data were sourced from the ERA5-land (Muñoz, 2019). The 8d leaf area index (LAI) and the 16d normalized difference vegetation index (NDVI) data, both with a spatial resolution of 500m, were downloaded from MODIS product of MOD15A2H (Myneni et al., 2021) and MOD13A1 (Didan, 2021), respectively.

**2.2 Hydrological model: THREW**

The hydrological model employed in this study is the Tsinghua Hydrological Model based on Representative Elementary Watershed (THREW) developed by Tian et al. (2006). THREW integrates a set of equilibrium equations for mass, momentum, energy, and entropy, along with constitutive relationships governing various fluxes between representative units and sub-regions within units. In THREW model, the REW is separated into two layers, i.e., surface layer and subsurface layer. Six sub-regions (or zones), i.e., bare soil zone (b-zone), vegetated zone (v-zone), snow covered zone (n-zone), glacier covered zone (g-zone), sub-stream-network (t-zone), and main channel reach (r-zone), are defined within the surface layer, and two sub-regions, i.e., unsaturated zone (u-zone) and saturated zone (s-zone), are defined within the sub-surface layer (Fig. 2). In such a way the principal landscape types can be explicitly treated in the REW approach (Tian et al., 2008). The

primary parameters utilized and to be calibrated in the THREW model are presented in Table 2. This model has demonstrated versatility across watersheds with diverse climates and geological conditions, such as Han River basin (Sun et al., 2014; Li et al., 2018), where many basins in this study are located, and other basins such as Yarlung Tsangpo-Brahmaputra River basin (Xu et al.,2019; Nan et al., 2021; Cui et al., 2023), Urumqi River basin (Mou et al., 2009).

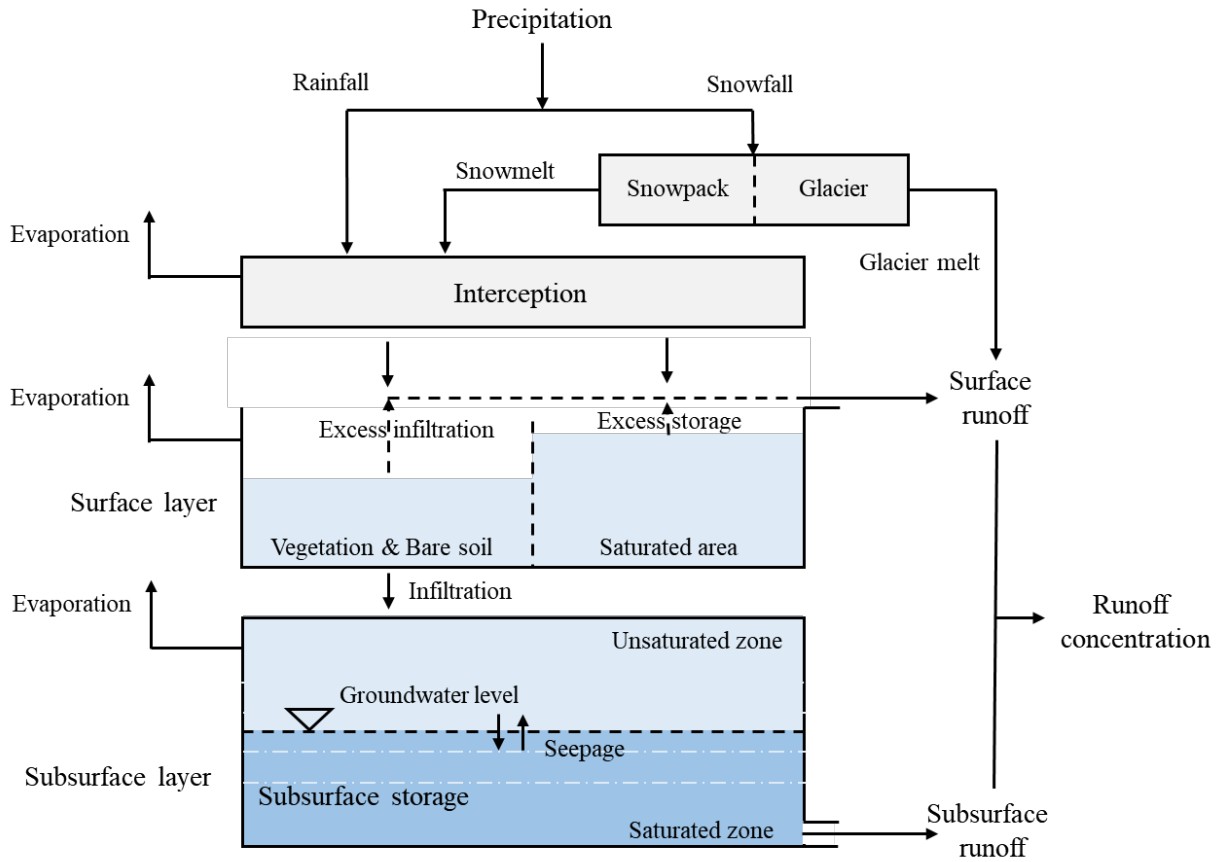

**Figure 2. Structural diagram of runoff generation processes in the THREW model**

**Table 2. Parameters to be calibrated in the THREW model.**

| Symbol | Unit | Physical descriptions | Range |
|---|---|---|---|
| Kv | - | fraction of potential transpiration rate over potential evaporation | 0-0.8 |
| nt | - | Manning roughness coefficient for hillslope | 0-0.2 |
| GaIFL | - | coefficient for spatial averaged infiltration capacity | 0-0.7 |
| GaEFL | - | coefficient for spatial averaged exfiltration capacity | 0-0.7 |
| WM | cm | Tension water storage capacity, which was used in the Xinanjiang model (Zhao et al., 1992) to calculate saturation area | 0-10 |
| B | - | Shape coefficient used in Xinanjiang model to calculate saturation area | 0-1 |
| KKA | - | Coefficient to calculate subsurface runoff in $R_g = KKD \cdot S \cdot K_S^S \cdot (y_s/Z)^{KKA}$, where S is the topographic slope, $K_S^S$ is the saturated hydraulic conductivity, $y_s$ is the depth of saturated ground- water and Z is the total soil depth | 0-6 |
| KKD | - | See description for KKA | 0-0.5 |

| | | | |
|---|---|---|---|
| $C_1$ | - | Coefficient to calculate the runoff concentration process using the Muskingum method: $Q_2 = C_1 \cdot I_1 + C_2 \cdot I_2 + C_3 \cdot Q_1 + C_4 \cdot Q_{lat}$, where $I_1$ and $Q_1$ are the inflow and outflow at the prior step, respectively; $I_2$ and $Q_2$ are the inflow and outflow at the current step, respectively; $Q_{lat}$ is lateral flow of the river channel; $C_3 = 1 - C_1 - C_2$; and $C_4 = C_1 + C_2$ | 0-1 |
| $C_2$ | - | See description for $C_1$ | 0-1 |

## 2.3 Model experimental design

To examine the influence of rainfall and runoff data at different resolutions on streamflow simulation, we designed two specific experiments: the daily test and the hourly test. The flowchart of the experimental tests is shown in Fig. 3, which is
120 described as follows:

(1) Daily test: This experiment was designed to investigate the impact of high-resolution rainfall data on daily streamflow simulation. The model was driven by rainfall data at various resolutions and calibrated using daily resolution streamflow data. This approach allowed us to assess whether (and to what extent) sub-daily rainfall data can enhance daily streamflow simulation, addressing the first question raised in the introduction.

(2) Hourly test: This experiment was designed to investigate the impact of high-resolution rainfall and streamflow data on hourly streamflow simulation. In this test, the temporal resolutions of rainfall data and measured streamflow data for calibration were the same, both set as various resolutions. The model was calibrated using streamflow data with the given temporal resolution, and then the hourly streamflow simulated by the calibrated model was evaluated based on the hourly measured data. This setup aimed to determine the necessary data resolution for providing reliable hourly streamflow
forecasts, thereby addressing the second question introduced earlier.

These experiments were designed to explore how data resolution impacts the accuracy and reliability of model's performance at different temporal scales. The computational time step of the model is also one of the factors impacting model's performance (Jie et al., 2017; Reynolds et al. 2017). To minimize any bias associated with varying computational time steps, the computational time step of hydrological simulation was consistently set at one hour. This standardization was
135 maintained regardless of the actual temporal resolution of the input data. All input data besides rainfall were resampled to the hourly resolution through averaging prior to simulation. As a result, the output, specifically the simulated runoff data, was consistently produced at an hourly scale.

In the daily test, rainfall data at various temporal resolutions was fed into the hydrological model to generate simulated hourly streamflow. This hourly data was then aggregated to daily scale for direct comparison with observed daily streamflow
data. Through this process, model parameters were calibrated by aligning the simulated outcomes with observed data, thus optimizing the parameters for different resolutions of rainfall data. An automatic optimization algorithm, Python Surrogate Optimization Toolbox (pySOT, Eriksson et al., 2019) was employed for model calibration, with the objective of maximizing Kling-Gupta Efficiency (KGE). The pySOT algorithm utilizes radial basis functions (RBFs) as surrogate models to approximate simulations, thereby reducing the runtime for each model iteration. During a single optimization process, the

optimization algorithm iteratively generates new parameters via the Symmetric Latin Hypercube Design (SLHD) method. The optimization ceases when the optimization objective converges or the number of iterations reaches a predetermined threshold (set at 3000). In this study, the pySOT algorithm was repeated 100 times, and the final parameter set was determined based on the optimal objective (maximum KGE). After calibration, another performance metric, Relative Error of Peak Flow (REP), was calculated. These metrics were calculated as follows:

$$KGE = 1 - \sqrt{(r-1)^2 + (\alpha-1)^2 + (\beta-1)^2} \tag{1}$$

$$REP = \frac{Q_{sim,p} - Q_{obs,p}}{Q_{obs,p}} \tag{2}$$

where, $r$ represents the Pearson correlation coefficient between simulated and observed values, $\alpha$ is the ratio of the mean of simulated values to the mean of observed values, $\beta$ is the ratio of the standard deviation of simulated values to the standard deviation of observed values, $Q_{sim,p}$ and $Q_{obs,p}$ are the simulated and observed peak flow, respectively.

The hourly test followed a similar procedure to the daily test, inputting rainfall data at various temporal resolutions into the hydrological model to produce simulated hourly streamflow. This output was aggregated to match the resolution of the input data and compared with the corresponding observed data for calibration. The performance of calibrated model on simulating hourly streamflow was then assessed by calculating KGE and REP, based on the hourly simulated and observed streamflow data.

The flowchart of the experimental tests was illustrated in Fig.3, where D and H refer to daily and hourly test, $x_i$ is each member of the time step (t.s.) set (TS), which consists of 1h, 2h, 3h, 4h, 6h, 12h and 24h. $KGE_{D,xi}$ and $REP_{D,xi}$ are the KGE of and REP of daily streamflow forced by rainfall at time step of $x_i$. Similarly, $KGE_{H,xi}$ and $REP_{H,xi}$ denote the KGE and REP for hourly streamflow at time step of $x_i$. To quantify the different model abilities on streamflow simulations at hourly and daily scales, rKGE and ΔREP were calculated as equations 3 and 4. Generally, rKGE is greater than 1 and ΔREP is positive because it is easier for models to achieve good performance on the streamflow simulation at coarse temporal resolutions. A rKGE close to 1 indicates that the model performs equally well at the hourly scale as it does at the daily scale, suggesting its applicability across different temporal scales and providing comparable reliability. A ΔREP approaching 0 has a similar implication. In addition to the metric calculated for different temporal resolutions, The average KGE and REP in the daily and hourly test, aggregating these metrics across temporal scales, were calculated to provide an average performance measure for the model within each catchment. The average KGE in the daily test and hourly test were calculated as equation 5 and 6, and the average relative error $REP_{D,ave}$ and $REP_{H,ave}$ were calculated in a similar way. The average rKGE and ΔREP were calculated as equations 7 and 8.

$$rKGE_{xi} = \frac{KGE_{D,xi}}{KGE_{H,xi}} \tag{3}$$

$$\Delta REP_{xi} = REP_{D,xi} - REP_{H,xi} \tag{4}$$

$$KGE_{D,ave} = average\{KGE_{D,xi}\} = \frac{\sum KGE_{D,xi}}{8} \quad (xi \in TS) \tag{5}$$

$$KGE_{H,ave} = average\{KGE_{H,xi}\} = \frac{\sum KGE_{D,xi}}{8} \quad (xi \in TS) \tag{6}$$

$$rKGE_{ave} = \frac{KGE_{D,ave}}{KGE_{H,ave}} \tag{7}$$

$$\Delta REP_{ave} = REP_{D,ave} - REP_{H,ave} \tag{8}$$

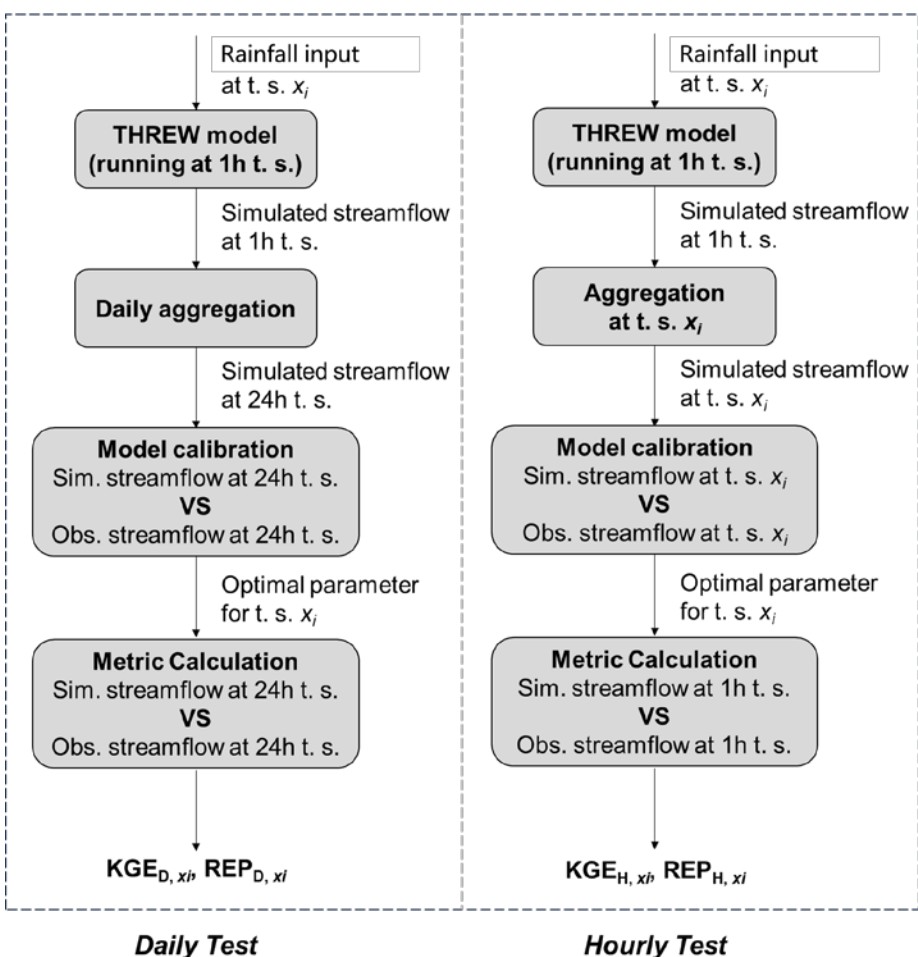

**Figure 3: Flowchart of the experimental tests**

### 2.4 Quantifying the value of high-resolution data

The paired two-sample t-test, a widely used statistical method to determine whether the means of two related groups of samples are significantly different (e.g., Xu et al., 2017), was adopted to test whether the performance of the hydrological model based on high-resolution data was significantly improved. Furthermore, the improvement index (IMP) was proposed to quantify this performance improvement. The IMP was calculated as follows:

$$IMP_D = \frac{Max\left\{KGE_{D,xi}\right\}}{KGE_{D,24h}} \tag{9}$$

$$IMP_H = \frac{Max\left\{KGE_{H,xi}\right\}}{KGE_{H,24h}} \tag{10}$$

$IMP_D$ and $IMP_H$ represent the highest degree of improvement of the sub-daily scale data relative to the daily scale data in the daily test and the hourly test, respectively. It should be noted that although IMP and rKGE have similar equations, they reflect different issues: the rKGE represents the difference in model ability on streamflow simulation at daily and hourly scale, while the IMP indexes quantify the value of sub-daily scale dataset on hydrological models.

## 2.5 Analyzing the influence factors of model performances

In order to identify potential factors affecting the performance of the model and determine which catchments benefit greater from high-resolution data, correlation analyses were performed between catchment attributes and model performance metrics. The model performance metrics were described in the previous section. The following catchment attributes were selected as potential factors: 1) drainage area (DRA); 2) mean annual rainfall (MAR); 3) mean annual runoff (MAQ); 4) runoff coefficient (QR); 5) goodness of uniform estimation (GOUE) for streamflow, which is the Nash-Sutcliffe efficiency coefficient (NSE) of hourly streamflow based on daily streamflow assuming a uniform intraday streamflow, revealing the intraday streamflow variation (Andréassian et al., 2001). The equation for the GOUE is shown as Eq. (11), where n is the length of the time series of the streamflow, $Q^h$ is the actual hourly streamflow, $\overline{Q^h}$ is the mean of the $Q^h$, $Q^{Dh}$ is the hourly streamflow based on daily streamflow assuming a uniform intraday streamflow. A larger GOUE indicates a smaller difference between daily and hourly streamflow, and a smaller intraday streamflow variation; 6) intraday standard deviation (ISTD) of the streamflow relative to the average streamflow; 7) rainfall gauge area (RGA), which is the catchment area divided by the number of rainfall stations in the catchment. Subsequently, correlation analyses were conducted between evaluation metrics (including $KGE_{D,ave}$, $KGE_{H,ave}$, $rKGE_{ave}$, $REP_{D,ave}$, $REP_{H,ave}$, $\Delta REP_{ave}$, $IMP_D$, $IMP_H$) and the above seven potential influencing factors. Statistical summaries of these attributes across the study catchments are presented in Table 1.

$$GOUE = 1 - \frac{\sum_{i=1}^{n}\left(Q_i^{Dh} - Q_i^h\right)^2}{\sum_{i=1}^{n}\left(Q_i^h - \overline{Q^h}\right)^2} \tag{11}$$

## 3 Results

## 3.1 Model performance at different time scales

The results of the daily and hourly tests are showed in Fig. 4 and detailed in Table 3. Considering the performance metrics obtained by various resolutions (ranging from 1h-24h) of data, in the daily test, the average KGE varied in the range of 0.84 - 0.87. The model performed worst when using the 24-hour resolution data, but even so, the KGE exceeded 0.8 in over 76%

of the catchments. At the 1-hour resolution, this proportion raised to 86%. As for REP, its average value at various data resolutions ranged between -16% and -24% indicating the general underestimation on peak flow. In the hourly test, the metrics got slightly worse compared with the daily test, with average KGE 0.03 lower and average REP 10% larger than that of daily test. The average KGE at various data resolutions varied in the range of 0.79 - 0.84, and the model produced KGE higher than 0.8 in over 75% of the catchments when using 1-hour rainfall and streamflow data. The average REP varied in the range ranges between -24% and -38%. The average rKGE and ΔREP varied within the ranges of 1.04-1.07 and 7.8%-14%, respectively. Furthermore, with the increasing temporal resolution, both rKGE and ΔREP exhibited a decline trend, suggesting that the discrepancy in the model's performance between hourly and daily scales diminished as the data resolution improved. In brief, at all data resolutions in both daily and hourly tests, the average KGE was consistently greater than 0.8, and the absolute value of average bias was consistently lower than 25%, respectively.

Besides, considering the metrics aggregated across different temporal scales for each catchment, the spatial patterns of average KGE and REP were shown in the Fig. 5 and detailed in the last column in Table 3. In most catchments, the average KGE exceeded 0.8 in both the daily and hourly experiments. The average absolute value of REP was less than 20% in the daily test and less than 30% in the hourly test. These results demonstrated the high performance and reliability of the THREW model in these catchments, with high KGE and low REP.

The evaluation metrics improved when using input and calibration data with higher temporal resolutions. Particularly, there was an obvious improvement in model performance when transitioning from daily to sub-daily resolution. In the daily test, the average KGE and REP obtained under 1-hour data driving were 0.03 and 8% higher, respectively, compared to those obtained under 24-hour data driving. In hourly testing, the difference in KGE and REP between 1-hour data and 24-hour data driven results were 0.05 and 14%, respectively. Figure 4c and 4f shows that the differences between daily and hourly metrics got smaller when higher resolution input data was used. Transitioning from 24-hour to 1-hour intervals in input data resolution resulted in a decrease in rKGE from 1.07 to 1.04, and a decrease in ΔREP from 14.1% to 7.76%. This signified a close proximity in the model's performance between forecasting hourly and daily streamflow when utilizing 1-hour resolution data.

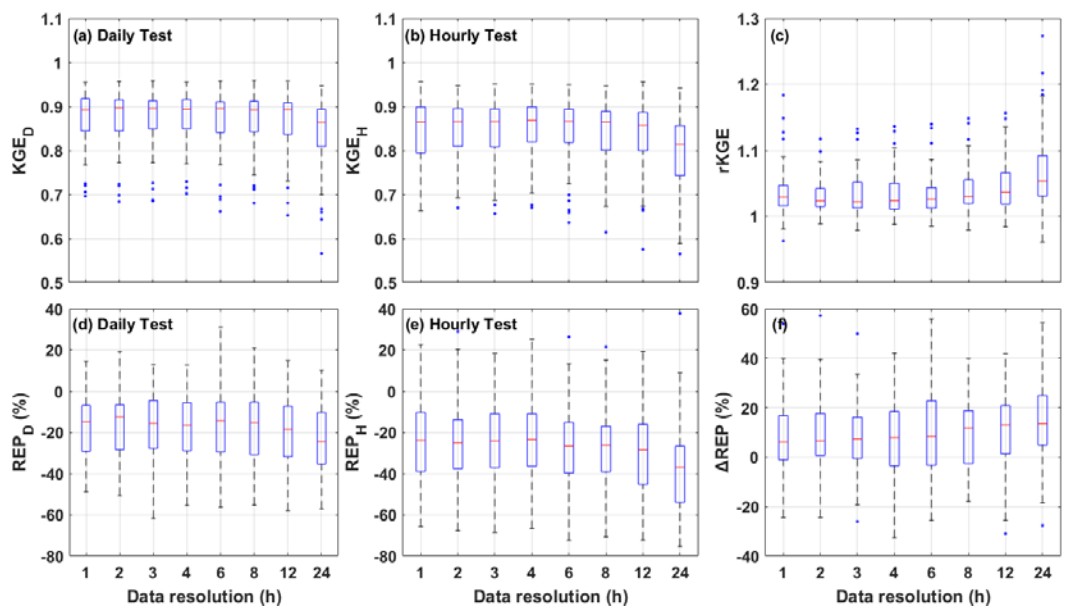

Figure 4: Box plot of KGE, REP, rKGE and ΔREP based on experiments across 63 catchments

Table 3. The characteristic values of KGE, REP, rKGE and ΔREP obtained by data with different temporal resolutions based on experiments across 63 catchments

| Metrics | Value | 1h | 2h | 3h | 4h | 6h | 8h | 12h | 24h | Average |
|---|---|---|---|---|---|---|---|---|---|---|
| | min | 0.696 | 0.683 | 0.685 | 0.702 | 0.661 | 0.680 | 0.653 | 0.565 | 0.668 |
| $KGE_D$ | max | 0.955 | 0.957 | 0.958 | 0.955 | 0.958 | 0.959 | 0.958 | 0.946 | 0.955 |
| | average | 0.873 | 0.872 | 0.874 | 0.873 | 0.871 | 0.871 | 0.867 | 0.840 | 0.867 |
| | min | 0.663 | 0.669 | 0.656 | 0.670 | 0.636 | 0.614 | 0.575 | 0.464 | 0.638 |
| $KGE_H$ | max | 0.956 | 0.947 | 0.950 | 0.950 | 0.949 | 0.946 | 0.956 | 0.942 | 0.946 |
| | average | 0.843 | 0.845 | 0.847 | 0.846 | 0.845 | 0.839 | 0.831 | 0.790 | 0.836 |
| | min | 0.963 | 0.989 | 0.979 | 0.988 | 0.985 | 0.979 | 0.984 | 0.961 | 0.981 |
| rKGE | max | 1.184 | 1.118 | 1.132 | 1.136 | 1.140 | 1.148 | 1.157 | 1.274 | 1.138 |
| | average | 1.037 | 1.033 | 1.032 | 1.033 | 1.032 | 1.040 | 1.047 | 1.068 | 1.040 |
| | min | -48.8 | -50.6 | -61.8 | -55.5 | -56.3 | -55.3 | -57.9 | -57.1 | -51.8 |
| $REP_D$ (%) | max | 43.0 | 54.2 | 56.2 | 54.2 | 31.1 | 21.0 | 14.8 | 10.2 | 32.8 |
| | average | -15.9 | -15.3 | -16.2 | -16.4 | -17.1 | -16.9 | -19.0 | -24.0 | -17.6 |
| | min | -65.6 | -67.8 | -68.5 | -66.6 | -72.3 | -70.7 | -72.3 | -75.4 | -69.0 |
| $REP_H$ (%) | max | 22.6 | 29.0 | 18.3 | 25.3 | 26.3 | 21.5 | 19.2 | 37.8 | 18.0 |
| | average | -23.7 | -23.8 | -24.3 | -23.4 | -26.5 | -26.7 | -30.9 | -38.1 | -27.2 |
| ΔREP | min | -24.4 | -24.4 | -26.0 | -32.6 | -25.6 | -18.0 | -31.0 | -27.6 | -19.6 |
| | max | 53.9 | 57.2 | 49.9 | 42.0 | 55.9 | 39.8 | 41.7 | 54.4 | 38.5 |
| (%) | average | 7.76 | 8.48 | 8.10 | 7.03 | 9.32 | 9.77 | 11.8 | 14.1 | 9.55 |

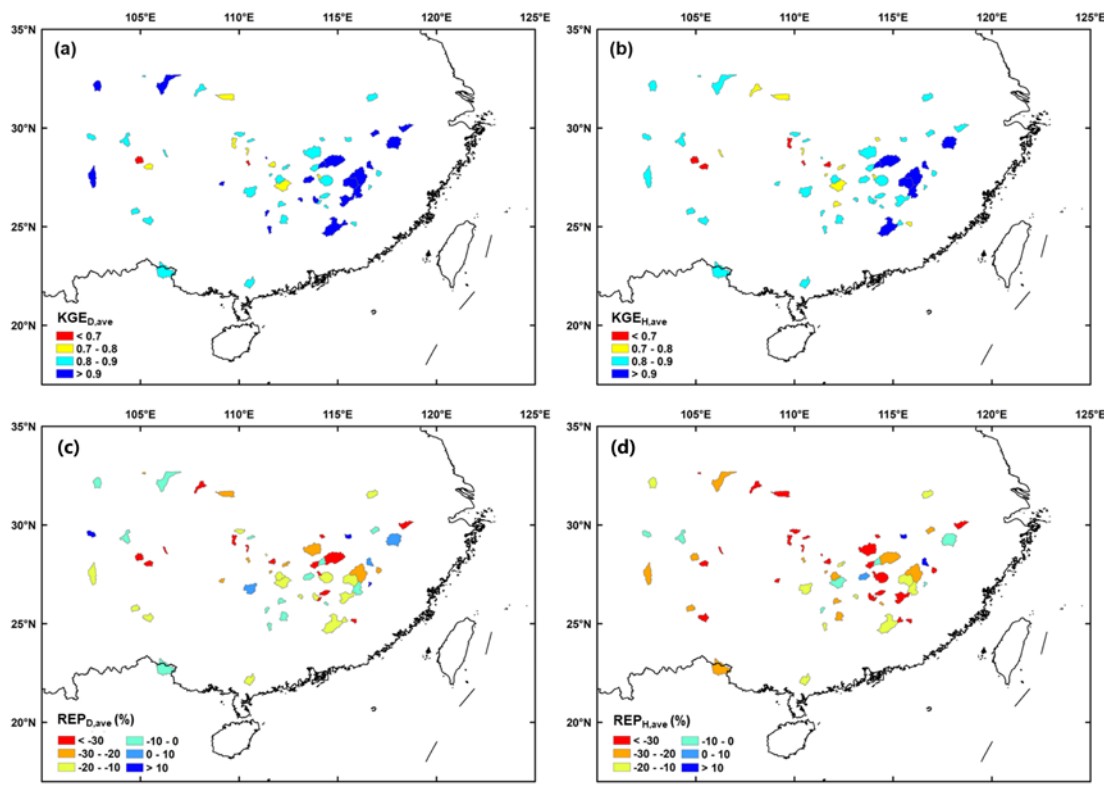

**Figure 5: Spatial pattern of KGE and REP**

## 3.2 Influence of data temporal resolution on model performance

In both the daily and hourly test, there was an obvious improvement in model performance when the rainfall/streamflow. For instance, in the daily test, when the data resolution shifted from 24h to sub-daily 12h, the average KGE increased from 0.84 to 0.87 and the average REP decreased from -24% to -19%. But such improvement got increasingly limited as the resolution further increased (Fig. 4). To quantify the difference in the model performances when adopting data with different resolutions, paired two-sample t-tests were conducted, and the results are shown in Table 4. In the daily test, significant improvement (at 0.01 significance level) on streamflow simulation was brought by sub-daily resolution rainfall data compared to the daily data, as indicated by the low p values in the last row of Table 4a and Table 4b. At least at the 0.05 significance level, there were significant differences in KGE obtained at 6-hour and 8-hour resolutions compared to that obtained at 12-hour resolution, but the difference in REP was not significant. The KGE obtained at 6-hour and 3-hour resolutions also showed significant differences, but the p-value was very close to 0.05. Meanwhile, it is noteworthy that the differences among evaluation metrics were insignificant at 0.05 level when the resolution of rainfall data was higher than 8-hour. This suggested that for daily streamflow forecasting, continuously increasing rainfall data resolution beyond the 8-hour threshold did not bring significant improvement on model performance. That is, the simulated daily streamflow obtained

from a model driven by 8-hour rainfall input had comparable reliability to that forced by 1-hour data, and effect of rainfall data with a temporal resolution exceeding 8 hours on enhancing daily forecasted flow was negligible.

Similar results were observed in the hourly test (Table 4c and Table 4d). Compared to resolutions of 24-hour and 12-hour,
utilizing higher-resolution data effectively enhanced the model's forecasting performance for hourly streamflow. Taking KGE as the performance metric, significant differences existed between the performance of the model when using 8-hour resolution data and that when using 2-6 hour resolution data. Notably, there were no statistical difference between the KGEs obtained by 8-hour and 1-hour resolutions, indicating that there was a decrease in KGE when the resolution is increased to 1-hour from 2-6 hours. Yet, from the perspective of average KGE (Fig. 4), this decrease was very slight, at most reaching
0.004. The difference in REP obtained by the 3-hour and 6-hour resolutions compared to that obtained by 8-hour resolution was not significant. When the temporal resolution of data used for forcing and calibrating models exceeded 6 hours, there was no significant improvement in the model's performance. Although the REP obtained at 6-hour and 4-hour resolutions exhibited significant differences, the p-value was very close to 0.05. Consequently, simulated hourly streamflow obtained by a model driven and calibrated by 6-hour data was comparably accurate to that driven and calibrated with 1-hour data, and
there was no significant added value brought by data with higher resolution than 6-hour.

**Table 4. P-values of the paired two-sample t-tests for each metric.**

**Table 4a P-values of the paired two-sample t-tests for KGE in daily test.** Bold values indicate $p < 0.05$, with * for $0.01 \leq p < 0.05$ and ** for $p < 0.01$.

| Resolution | 1h | 2h | 3h | 4h | 6h | 8h | 12h |
|---|---|---|---|---|---|---|---|
| 2h | 0.6277 | | | | | | |
| 3h | 0.5988 | 0.3039 | | | | | |
| 4h | 0.8525 | 0.6220 | 0.3443 | | | | |
| 6h | 0.3437 | 0.6170 | **0.0486*** | 0.2234 | | | |
| 8h | 0.3658 | 0.6131 | 0.1148 | 0.269 | 0.9369 | | |
| 12h | **0.0328*** | **0.0300*** | **0.0011**** | **0.0025**** | **0.0119**** | **0.0086**** | |
| 24h | **2.8E-09**** | **2.5E-09**** | **4.2E-11**** | **1.2E-10**** | **9.3E-12**** | **4.8E-11**** | **6.9E-12**** |

**Table 4b P-values of the paired two-sample t-tests for REP in daily test.** Bold values indicate $p < 0.05$, with * for $0.01 \leq p < 0.05$ and
275 ** for $p < 0.01$.

| Resolution | 1h | 2h | 3h | 4h | 6h | 8h | 12h |
|---|---|---|---|---|---|---|---|
| 2h | 0.4736 | | | | | | |
| 3h | 0.7260 | 0.2896 | | | | | |
| 4h | 0.5653 | 0.1686 | 0.789 | | | | |
| 6h | 0.2874 | 0.1106 | 0.3414 | 0.5084 | | | |
| 8h | 0.2659 | 0.1327 | 0.4602 | 0.5592 | 0.8511 | | |
| 12h | **0.0132*** | **0.0033** | **0.0205*** | **0.0305*** | 0.0909 | 0.0593 | |
| 24h | **2.8E-06**** | **1.3E-06**** | **8.8E-06**** | **7.7E-06**** | **2.3E-07**** | **3.6E-06**** | **0.0006**** |

**Table 4c P-values of the paired two-sample t-tests for KGE in hourly test.** Bold values indicate p < 0.05, with * for 0.01 ≤ p < 0.05 and ** for p < 0.01.

| Resolution | 1h | 2h | 3h | 4h | 6h | 8h | 12h |
|---|---|---|---|---|---|---|---|
| 2h | 0.3494 | | | | | | |
| 3h | 0.0931 | 0.3611 | | | | | |
| 4h | 0.3350 | 0.7743 | 0.5075 | | | | |
| 6h | 0.4749 | 0.9775 | 0.3646 | 0.8304 | | | |
| 8h | 0.1511 | **0.0202\*** | **0.0034\*\*** | **0.0057\*\*** | **0.0011\*\*** | | |
| 12h | **0.0004\*\*** | **0.0001\*\*** | **3.2E-06\*\*** | **1.2E-05\*\*** | **6.3E-07\*\*** | **0.0002\*\*** | |
| 24h | **6.7E-12\*\*** | **7.6E-13\*\*** | **1.9E-13\*\*** | **9.4E-13\*\*** | **1.5E-14\*\*** | **2.4E-13\*\*** | **7.6E-16\*\*** |

**Table 4d P-values of the paired two-sample t-tests for REP in hourly test.** Bold values indicate p < 0.05, with * for 0.01 ≤ p < 0.05 and ** for p < 0.01.

| Resolution | 1h | 2h | 3h | 4h | 6h | 8h | 12h |
|---|---|---|---|---|---|---|---|
| 2h | 0.9048 | | | | | | |
| 3h | 0.6343 | 0.6620 | | | | | |
| 4h | 0.8370 | 0.7715 | 0.5732 | | | | |
| 6h | 0.0637 | 0.0556 | 0.1390 | **0.0429\*** | | | |
| 8h | **0.0364\*** | **0.0364\*** | 0.1023 3 | **0.0312\*** | 0.8301 | | |
| 12h | **4.6E-06\*\*** | **6.2E-06\*\*** | **6.4E-06\*\*** | **3.1E-05\*\*** | **0.0009\*\*** | **0.0011\*\*** | |
| 24h | **1.5E-09\*\*** | **2.2E-10\*\*** | **9.6E-10\*\*** | **1.4E-08\*\*** | **5.7E-09\*\*** | **2.5E-08\*\*** | **1.1E-05\*\*** |

To further quantify the benefits from higher resolution data to hydrological simulation, the IMP indexes were calculated and showed in Fig. 6. In the daily test, the improvement in daily KGE brought by sub-daily rainfall input data ranged from -1.6% to 24.1% (5.3% on average). Specifically, higher resolution rainfall data caused slight decrease in daily streamflow simulation efficiency in two catchments, and the increase in daily KGE was lower than 5% in more than half of the catchments (40 of 63). In the hourly test, the improvement brought from sub-daily resolution data was significantly

correlated to that in the daily test ($R^2$=0.72, p<0.01), and the improvement was more pronounced. The model performance was improved by the sub-daily rainfall and streamflow data in all the catchments, with the increase in hourly KGE ranging from 0.1% to 52.9% (9.3% on average). The IMP had no significant spatial distribution pattern, but it was likely larger in smaller catchments (also showed in Fig. 7).

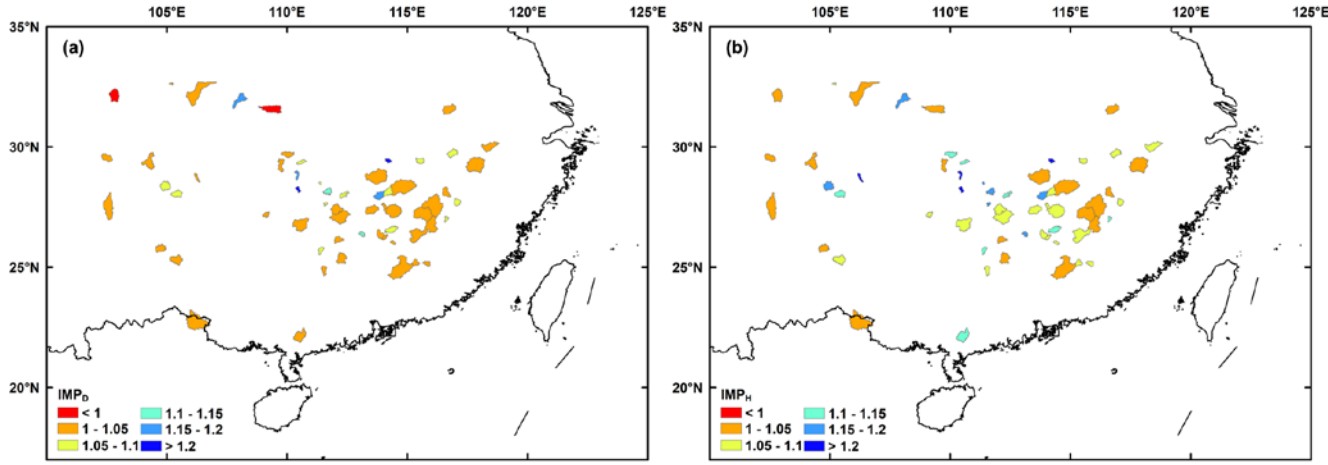

Figure 6: Spatial pattern of the benefit from higher resolution data. (a) IMP in daily test; (b) IMP in hourly test

## 3.3 Influence factors of model performance and the improvement brought by high-resolution data

In order to identify the potential factors affecting model performance and improvement, correlation analysis was carried out between catchment attributes and model performance metrics, and the results are shown in Table 5. Overall, the GOUE was the strongest predictor among the influence factors with highest correlation coefficient for most model performance and improvement metrics, except for the two metrics evaluating peak simulation. In both daily and hourly tests, the KGE was significantly positively correlated with DRA, MAR, and GOUE (detailed in the Fig. 7), indicating that the model performed better on daily and hourly streamflow simulations in catchments with larger drainage area, more rainfall and higher goodness of uniform estimation (lower intraday streamflow variation). This could be attributed to the general fact that larger watershed areas tend to exhibit relatively stable and slower changes in streamflow, and the generally stronger correlation between rainfall and runoff in catchments with wetter conditions resulting in more predictable streamflow. In addition, the KGE for hourly streamflow was positively correlated with MAQ (r=0.279, p<0.05) and negatively correlated with ISTD (r=-0.260, p<0.05), indicating that the model performed better in catchments with larger runoff and less intraday streamflow variation. In arid regions and areas with significant diurnal flow fluctuations, streamflow forecasting at fine temporal scales poses greater challenges. REP was significantly negatively correlated with ISTD in both daily (r=-0.272, p=0.031) and hourly scale (r=-0.324, p<0.01), indicating that the model performed better on peak flow simulation in catchments with less intraday streamflow variation (Fig. 8). Predicting peaks accurately is more challenging in data sequences with larger fluctuations. rKGE was negatively correlated with DRA (r=-0.393, p<0.01) and GOUE (r=-0.672, p<0.01), and positively correlated with ISTD (r=0.350, p<0.01), indicating that the model exhibited a closer performance between daily and hourly scales in catchments with larger drainage area and lower diurnal streamflow variability (Fig. 9).

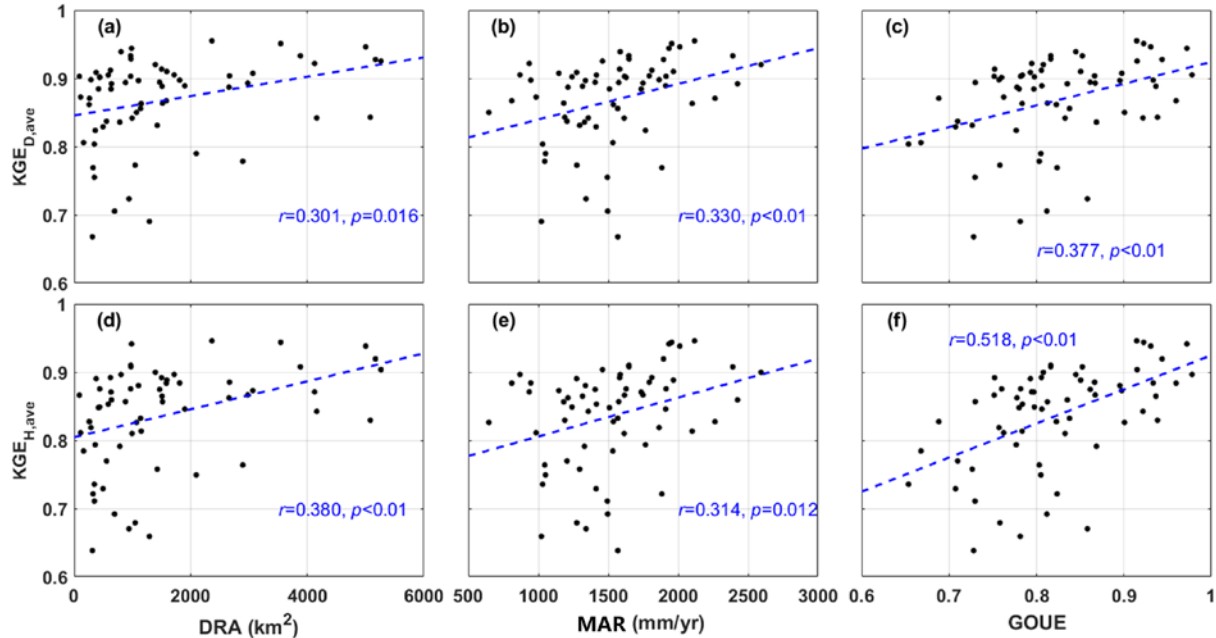

**Figure 7: Scatter diagram of KGE and impact factors (DRA, MAR, GOUE)**

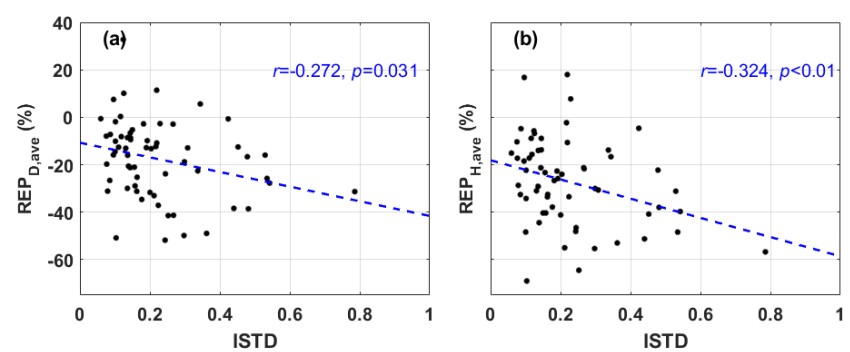

**Figure 8: Scatter diagram of REP and ISTD**

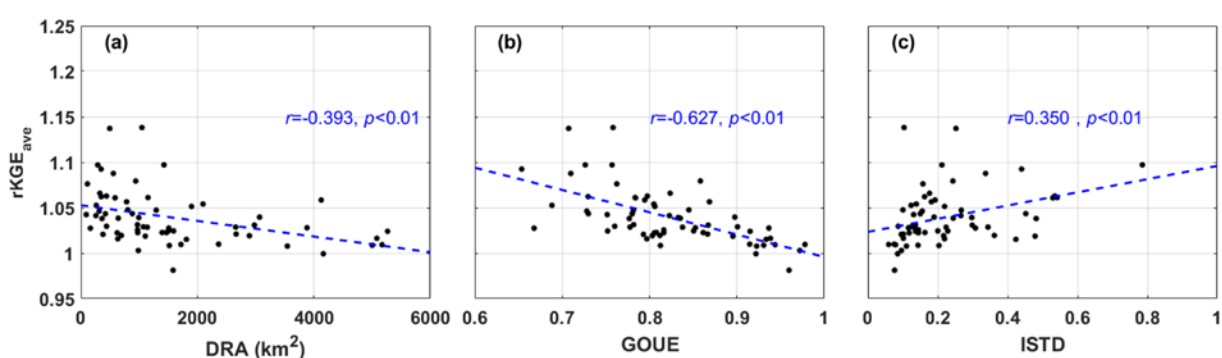

**Figure 9: Scatter diagram of rKGE and DRA, GOUE, ISTD**

Figure 10 shows the relationship between IMP and the significant influence factors. First, the IMP was negatively correlated with DRA (r=-0.408 and -0.490 for daily and hourly test, p<0.01 for both), indicating that the benefit of sub-daily rainfall and streamflow data was more significant in smaller catchments. This could be attributed to the fast rainfall-runoff response in smaller catchments leading to sharp streamflow variations. The model forced and calibrated by the higher temporal resolution data contained more fine-grained information, consequently being more likely to produce a realistic description of hydrological processes and capture the streamflow variations. Second, the IMP was negatively correlated with GOUE (r=-0.489 and -0.643 for daily and hourly test, p<0.01 for both), and the reason was similar with that of the negative relation between IMP and DRA. Last, the IMP was negatively correlated with RGA (r=-0.295, p=0.020 and r=-0.330, p<0.01 for daily and hourly test), indicating that the benefit from higher resolution data was more significant in catchments with more rain stations, which could provide more detailed and refined rainfall input data and consequently improve model performance.

**Table 5. Correlation coefficient among evaluation metrics and catchment attributions.** Bold values indicate p < 0.05, with * for 0.01 ≤ p < 0.05 and ** for p < 0.01.

| | $KGE_{D,ave}$ | $KGE_{H,ave}$ | $rKGE_{ave}$ | $REP_{D,ave}$ | $REP_{H,ave}$ | $\Delta REP_{ave}$ | $IMP_D$ | $IMP_H$ |
|---|---|---|---|---|---|---|---|---|
| DRA | 0.301* | 0.380** | -0.393** | 0.131 | 0.208 | -0.133 | -0.408** | -0.490** |
| MAR | 0.330** | 0.314* | -0.120 | 0.056 | 0.036 | 0.022 | 0.080 | 0.060 |
| MAQ | 0.247 | 0.279* | -0.230 | 0.105 | 0.123 | -0.040 | -0.141 | -0.023 |
| QR | -0.115 | -0.031 | -0.206 | 0.102 | 0.152 | -0.088 | -0.250* | -0.079 |
| GOUE | 0.377** | 0.518** | -0.627** | 0.214 | 0.304* | -0.163 | -0.489** | -0.643** |
| ISTD | -0.171 | -0.260* | 0.350** | -0.272* | -0.324** | 0.114 | 0.302* | 0.193 |
| RGA | 0.197 | 0.247 | -0.244 | 0.136 | 0.127 | -0.006 | -0.295* | -0.330** |

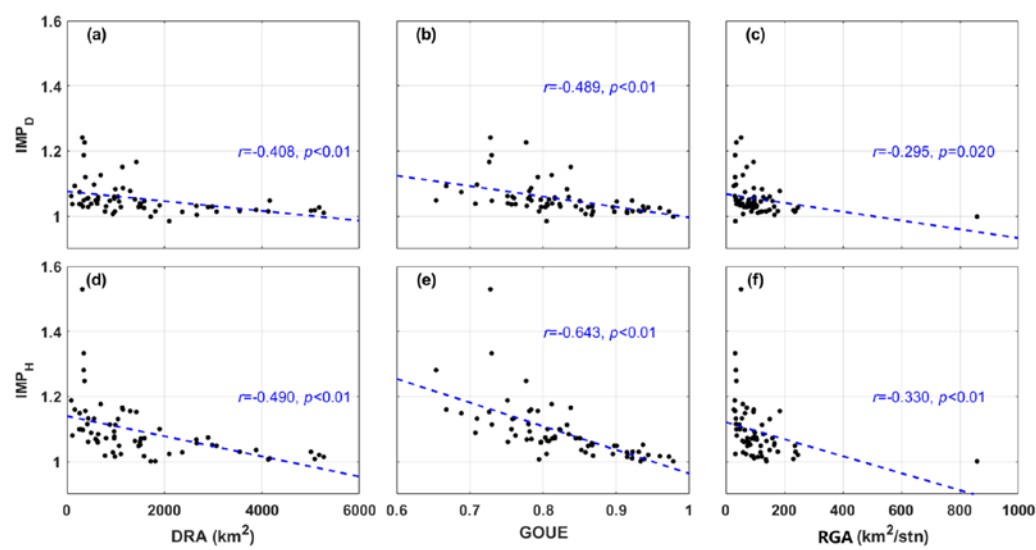

**Figure 10: Scatter diagram of IMP and impact factors (DRA, GOUE, RGA)**

### 3.4 Specific insights from representative cases

To further analyze how rainfall and streamflow data at different resolutions specifically influence the hydrological simulation results, we selected three representative catchments based on the Hourly Test results (IMP$_H$) and sensitive factors such as DRA, GOUE and RGA, which were identified as having a significant impact on IMP$_H$. These catchments were chosen as representative examples specifically due to their following typical characteristics and the distinct patterns they exhibit in hydrological simulation results, providing valuable insights into the influence of data resolution on model performance.

Catchment 1 (Tiantangyan): This catchment, characterized by relatively small DRA, GOUE, and RGA values, showed a significant improvement in simulation results with increased data resolution, as reflected by a large IMP.

Catchment 2 (Saitang): With medium values for DRA, GOUE, and RGA, this catchment demonstrated a gradual improvement in KGE as resolution increased, though the gains were less substantial.

Catchment 3 (Gaoan): As one of the largest catchments in terms of DRA, with relatively large GOUE and RGA, Gaoan exhibited limited improvement in performance with higher-resolution data, as indicated by a smaller IMP.

The attributes of these representative catchments are listed in Table 6 (catchment names correspond to the names of the hydrological stations at their respective outlets). The KGE values obtained from data of different resolutions are shown in Table 7.

**Table 6.  Attributes of representative catchments**

| ID | Name | DRA (km$^2$) | GOUE | RGA (km$^2$/stn) | IMP |
|---|---|---|---|---|---|
| 1 | Tiantangyan | 347 | 0.65 | 34.70 | 1.281 |
| 2 | Saitang | 2978 | 0.87 | 165.47 | 1.051 |
| 3 | Gaoan | 5172 | 0.94 | 246.28 | 1.021 |

**Table 7. KGE of each representative catchment obtained from data of different resolutions in the hourly test**

| Name | 1h | 2h | 3h | 4h | 6h | 8h | 12h | 24h |
|---|---|---|---|---|---|---|---|---|
| Tiantangyan | 0.776 | 0.781 | 0.777 | 0.75 | 0.755 | 0.73 | 0.706 | 0.61 |
| Saitang | 0.881 | 0.874 | 0.867 | 0.868 | 0.865 | 0.873 | 0.865 | 0.839 |
| Gaoan | 0.922 | 0.924 | 0.925 | 0.921 | 0.92 | 0.918 | 0.917 | 0.906 |

The hourly time series of rainfall, simulated flow, and observed flow for the representative catchments under 1-hour and 24-hour resolution data for the entire study period (2014-2015) are shown in Figure 11. Across all representative catchments and data resolutions, the model successfully simulated the majority of flood events and baseflow. Furthermore, to gain more specific and clear insights into the issue, the flood event with the highest peak flow during the study period was selected as the representative case, and the hydrological time series for one or three weeks before and after these events are shown in Figure 12. In particular, the results at a 6-hour resolution have been added to the figure, as this has been identified as a threshold resolution.

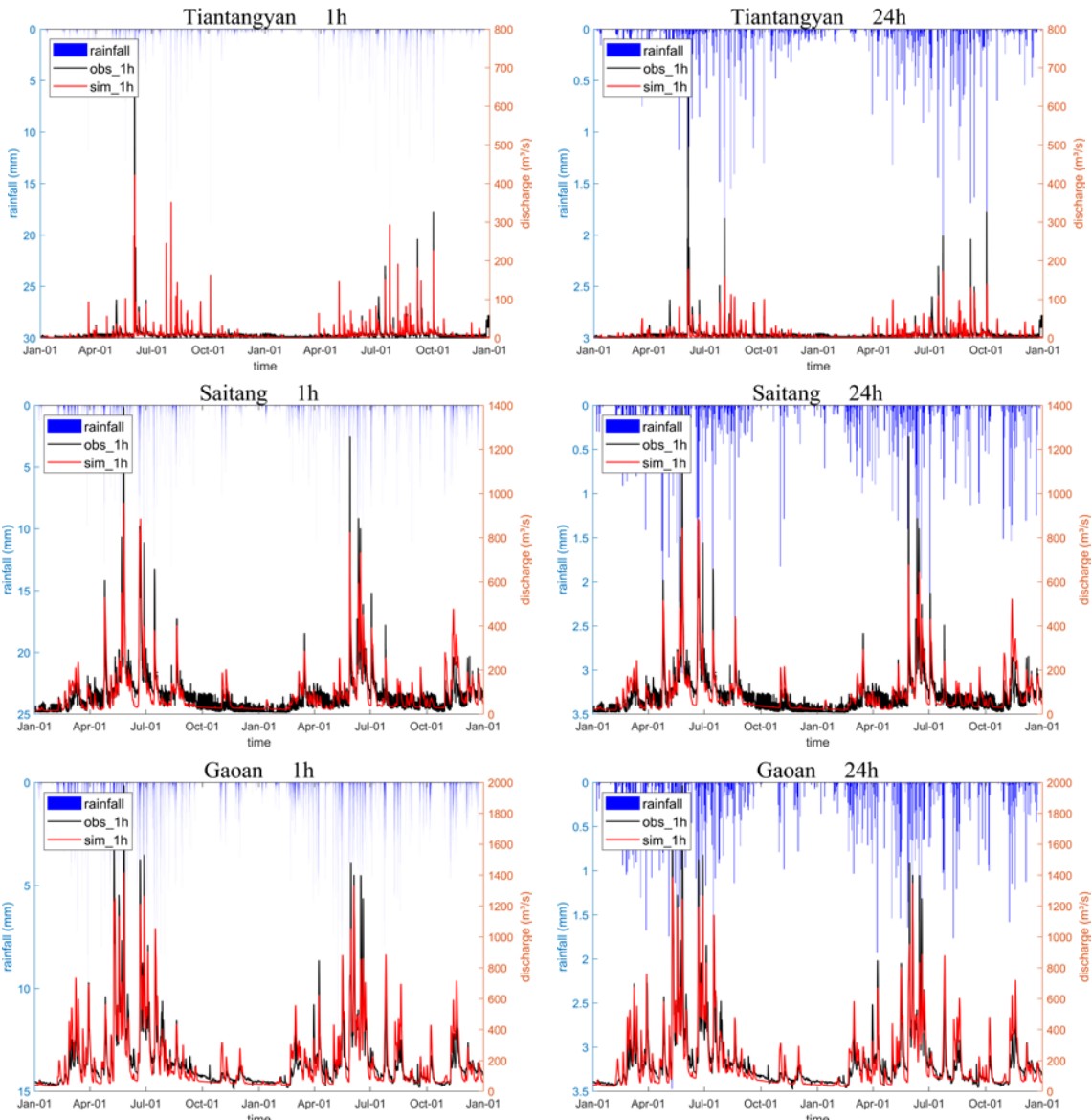

**Figure 11. Hourly time series of rainfall, simulated flow, and observed flow for the representative catchments under 1-hour and 24-hour resolution data for the entire study period (2014-2015).** Note that when the original data resolution is 24 hours, the rainfall is averaged and resampled to an hourly resolution.

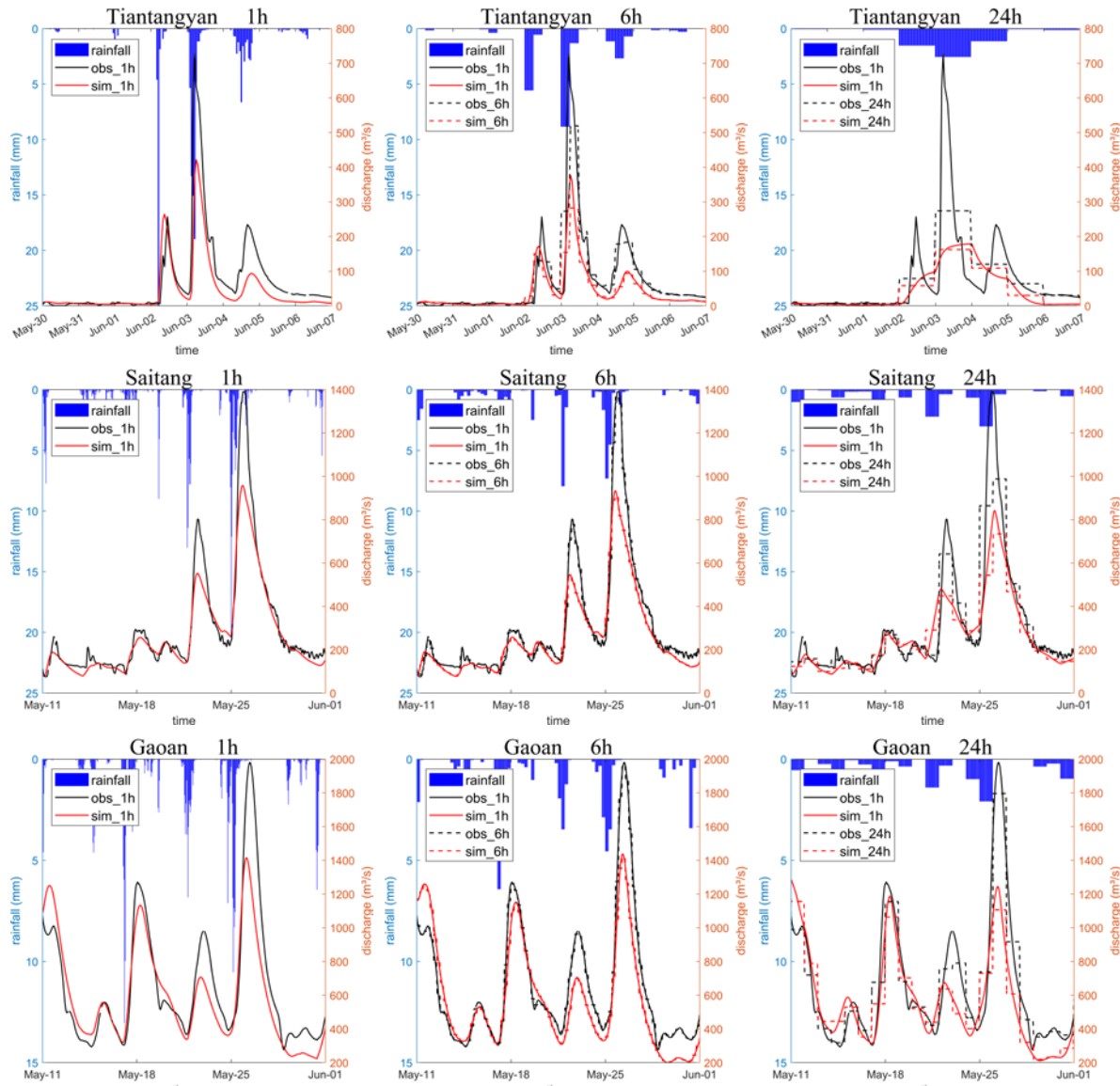

**Figure 12. Hourly time series of rainfall, simulated flow, and observed flow for the representative flood events under 1h, 6h and 24h resolutions.** The blue bars indicate the resampled hourly average precipitation for the catchment (used by the model during simulation). The red solid line indicates the hourly simulated flow output from the model. The red dashed line represents the simulated flow aggregated to other time resolutions (6h or 12h). The black solid line shows the actual hourly observed flow, and the black dashed

line represents the observed flow aggregated to other time resolutions.

In the case of the Tiantangyan catchment, data at a 1-hour resolution show that there were three distinct rainfall and corresponding flood events within the period. The streamflow changed rapidly, with each flood lasting less than one day, while floods in other basins typically span several days. When using a 24-hour resolution, it is difficult to distinguish the three independent rainfall events, resulting in a single flood event lasting three days in the simulation. With a 6-hour

resolution, despite some peak discrepancies, both the rainfall and observed streamflow data closely capture the real rainfall-

runoff process, significantly improving simulation accuracy. At a 1-hour resolution, there is some improvement in peak simulation for the first flood event, with no notable changes in other events.

In the case of the Saitang catchment, the observed flow data at a 24-hour resolution have large errors compared to the actual 1-hour resolution, whereas the measured flow at 6-hour and 1-hour resolutions are very close. Thus, improving the resolution from 24 hours to 6 hours enhanced simulation accuracy significantly, but further increasing it to 1 hour offered no substantial improvement, as no additional effective information is introduced.

In the case of the Gaoan catchment, the streamflow changes more smoothly in this larger catchment. The observed flow at a 24-hour resolution already closely approximates actual flow, with the relative error in peak flow being only -8.8%, compared to -62.3% and -28.9% for the peak flow errors in the other two catchments. Furthermore, when the resolution of the observed flow is increased to 6-hour, it exhibits negligible discrepancy when compared to the data at 1-hour resolution. Thus, when using 24-hour resolution data, the model performs well on an hourly scale, and continuous improvement in resolution did not lead to a notable increase in simulation accuracy.

# 4 Discussions

## 4.1 Implication on the measurement of rainfall and streamflow

This study assessed the value of high-resolution rainfall and streamflow data on hydrological modelling. In terms of streamflow simulation at daily scale, results indicated that increasing the temporal resolution of rainfall data from daily to sub-daily intervals significantly improved the simulation accuracy, aligning with the general expectation of the role of high-resolution data in enhancing predictive performance. However, the improvement on daily streamflow simulation brought by rainfall input data with resolution higher than 8-hour was limited. The similar phenomenon was observed in the test on hourly streamflow simulation, and the resolution threshold above which higher resolution brought negligible improvement was 6-hour. These findings seemed to contradict intuitive expectations that higher resolution data consistently benefits hydrological models, but similar results were reported in previous articles across different catchments using various models. For instance, Ficchì et al. (2016) conducted a study in 240 French catchments, utilizing the GR4 rainfall-runoff model forced by rainfall inputs at eight different time steps ranging from 6 minutes to 1 day, to investigate the extent to which the performance of hydrological modeling is improved by short time-step data. Their conclusion is that, on average and within their set of catchments, using shorter model time steps provides no additional value for reference time steps equal to or shorter than 6 hours (when evaluating outputs aggregated at the reference time scales). Reynolds et al. (2017) examined the relationship between model performance and the transferability of parameter sets calibrated at various temporal resolutions in two small catchments in Central America, employing the HBV model. They found that parameters calibrated at the daily provided peak-flow simulations almost as good as parameters calibrated at sub-daily resolutions. Tudaji et al. (2024) investigated the impact of different input data resolutions using a four-source hydrological model in seven catchments in northern China, and found that for daily streamflow simulations, improvements in model performance become negligible

when the resolution exceeds 12 hours. As for hourly streamflow simulations, improvements in overall flood process accuracy become negligible when the resolution of input exceeds 6 hours, while higher resolutions further enhance the precision of peak flow simulations.

Although the involved catchments and models differed among different studies, the findings were fundamentally consistent: higher resolution data did not necessarily guarantee better model performance. There are several reasons that may constrain the additional benefits of higher resolution data. From a data-driven perspective, first and foremost, due to the spatial and temporal autocorrelation of data such as rainfall and runoff, further increases in resolution beyond a certain threshold may not necessarily yield additional valuable information. The fundamental value of high-resolution data lies in its capacity to introduce novel, accurate, and valuable information, which is essential for enhancing model performance. In this study, the GOUE metric was employed to partially quantify the additional information contributed by high-resolution data. Specifically, GOUE reflects the informational gain of actual hourly resolution data compared to data obtained by uniformly resampling daily resolution data to an hourly scale. Among various basin characteristics, GOUE exhibited the strongest correlation with IMP, suggesting that basins with greater streamflow variability benefit more from increased data resolution, as it introduces more detailed and novel information. This is further supported by the case studies of three representative basins, which vividly demonstrate that the ability of higher-resolution data to provide additional, accurate, and effective information is crucial for realizing its potential to enhance model performance. Secondly, the model's input data, particularly rainfall, might exhibit a lower signal-to-noise ratio at shorter time scales, owing to challenges in data validation and increased uncertainty in areal averaged rainfalls (Obled et al., 2009). Thirdly, specific to our study, we focused on the value of rainfall and streamflow data, while the resolution of other auxiliary data used in the model is fixed and relatively coarse. The resolution of other driving data, especially for the processes with significant diurnal variations, such as evapotranspiration (currently at a daily resolution), could be a factor constraining the model's ability to improve the model's performance at finer temporal scales.

From the perspective of model structure , the model itself may not be well-suited to capture the increased complexity of processes at shorter time steps. A real watershed acts like a low-pass filter, smoothing out short-term variability in input data and reducing the sensitivity of runoff to rapid changes (Ficchì et al., 2016). In contrast, hydrological models are simplified representations of actual runoff processes and often lack mechanisms to accurately replicate this natural filtering effect, such as dynamic re-infiltration or flow routing. When high temporal resolution data are used to drive these models, the simulated streamflow may exhibit excessive variability because the models directly reflect the rapid fluctuations present in the input data without the smoothing effects inherent in natural systems. This excessive variability can lead to discrepancies between simulated and observed streamflow, ultimately reducing the model's performance. For instance, localized short-duration heavy rainfall may cause infiltration-excess runoff, but during the surface flow routing process over the hillslope, re-infiltration can occur, leading to slower runoff variations (Li et al., 2022; Zhang et al., 2020). Although the hydrological model does not explicitly account for re-infiltration, the use of coarse-resolution data effectively performs a buffering function, which can result in comparable or even better performance than when using high-resolution data. Additionally,

Melsen et al. (2016) argued that the calibration and validation time interval should keep pace with the increase in spatial resolution in order to resolve the processes that are relevant at the applied spatial resolution. Some simple empirical formulas within the model may not be applicable at shorter time scales.

Considering the limited enhancement in model performance from high-resolution data, which varies across watersheds with different characteristics, this sheds light on the measurement of rainfall and streamflow. As stated by Seibert et al. (2024), "balancing the information content of additional measurements with their costs allows one to allocate resources efficiently, ensuring that you get your money's worth". When planning new monitoring facilities, it is essential to conduct a preliminary assessment of the value of current monitoring capabilities for building hydrological models and conducting hydrological

forecasts, based on the forecast objectives, precision requirements, and the accuracy of existing data. This assessment can verify the necessity of monitoring and determine the areas where monitoring efforts should be increased based on model and watershed characteristics. For instance, in our study catchments, the benefits of sub-daily resolution data were more pronounced in catchments with smaller drainage area, stronger intraday streamflow variation and more existed rainfall gauges. Thus, when conducting hourly hydrological forecasts using the THREW model in this region, it is advisable to

increase monitoring efforts and enhance monitoring frequency in catchments with smaller drainage areas or stronger intraday streamflow variation. However, in catchments where the temporal resolutions of rainfall and streamflow data are already higher than 6-hour, investing more costs in further increasing measurement resolution may not bring significant improvement to model efficiency. Additionally, the negative correlation between IMP and RGA also offers an implication that when upgrading the model, which based on coarse temporal resolution inputs to those with higher temporal resolution,

such as switching from daily to hourly data, concurrently increasing rain gauge density can help enhance model performance.

## 4.2 Limitations

Despite this study employed extensive catchment data and multiple evaluation metrics to derive general conclusions, some potential limitations warrant further consideration. First, this study adopted a specific model, and the selected catchments were mainly distributed in the southeastern China with similar climate types due to the data availability. The applicability of

465 the conclusions to other climatic regions and models requires further verification. Second, results showed that the benefit of high-resolution rainfall/streamflow data to daily and hourly streamflow simulation was negligible when the temporal resolution was higher than a threshold (8-hour and 6-hour for daily and hourly simulation), and the possible mechanism of such phenomenon has been primarily discussed. However, a deeper analysis and validation on such threshold effect are still lacking, which needs further investigation, such as comparative studies using distributed hydrological models with stronger

physical mechanisms and higher resolution. Last, the time of model running during calibration procedure was limited. Although it proved to be enough to produce a good simulation based on our previous modeling and calibration practice (e.g., Nan and Tian, 2024a, 2024b), it cannot ensure that the globally optimal result could be found. Consequently, it is difficult to determine whether the slightly decreasing model performance in some catchments is caused by high-resolution data or local optimal results.

## 5 Conclusions

This study evaluated the impact of high-resolution rainfall and streamflow data on hydrological modeling across 63 small-to-medium-scale catchments in Southeastern China. The models utilized data with resolutions ranging from 1 to 24 hours for both forcing and calibration, and the influence factors of model performance and improvements brought by high-resolution data were analyzed. Key findings are summarized as follows:

(1) Resolution impact on daily simulation: Increasing the resolution of the rainfall from daily (24h) to sub-daily (12h) resolution significantly enhanced daily streamflow forecasts, bringing 3% and 5% improvement to average KGE and peak flow simulation, respectively. However, rainfall input data with resolutions finer than 8 hours showed negligible improvement.

(2) Resolution impact on hourly simulation: While sub-daily rainfall and streamflow data significantly improved hourly streamflow simulation compared to daily data, improvements were minimal when the data resolution was finer than 6 hours. For the studied catchments, data with a 6-hour resolution generally provided adequate confidence for hourly simulations as hourly data.

(3) Model performance in varied conditions: Drainage area and intraday streamflow variation are significant influence factors on model performance and benefit of high-resolution data. In specify, the THREW model showed enhanced performance in catchments with larger drainage areas, higher rainfall, and less intraday streamflow variation. Sub-daily resolution data brought greater benefits in catchments with smaller drainage areas, more pronounced intraday flow variations, and a higher number of rain gauges.

*Competing interests.* At least one of the (co)-authors is a member of the editorial board of Hydrology and Earth System Sciences.

*Acknowledgements.* This study was made possible through the generous financial support from the Fund Program of State Key Laboratory of Hydroscience and Engineering under grant number sklhseTD-2024-C01 and the National Key Research and Development Program of China under grant number 2022YFC3002900. Additionally, the authors acknowledge the support from the National Natural Science Foundation of China under grant number 52309023 and the Shuimu Tsinghua Scholar Program.

*Code and data availability.* The data and the code of the THREW model used in this study are available by contacting the authors.

*Author contributions.* YN and FT conceived the idea; MT and YN conducted analysis; FT provided comments on the analysis; and all the authors contributed to writing and revisions.

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
