# Peer review of "Assessing the value of high-resolution rainfall and streamflow data for hydrological modeling: An analysis based on 63 catchments in southeast China"

_EGUsphere, 2024_

## Referee Comment (RC1)

This paper provides a valuable quantitative assessment of the impact of input data temporal resolution on hydrological modeling, offering significant insights for researchers and practitioners in this field. The study is both academically and practically relevant.

The authors conducted extensive watershed modeling and designed two experiments based on common forecasting tasks, using multiple evaluation metrics, which enhances the reliability of the conclusions. The paper successfully links its findings to existing literature, addressing a gap where similar conclusions required validation in different regions. By confirming these findings in a new area, the authors strengthen the reliability and generalizability of the conclusions, contributing valuable insights to the existing body of knowledge.

Overall, this paper is well-designed, content-rich, and makes a meaningful contribution to hydrological modeling. I recommend it for acceptance after revisions.

Although this paper is overall well-done, I believe the following points require further clarification and revision to help improve the paper's overall quality.

1. In addition to the rainfall and runoff data already mentioned in the paper, other data used for modeling should also be detailed.

2. It would be beneficial to reference Figure 2 at the beginning of Section 2.3 when introducing the experimental design. This will help readers refer to the flowchart while reading the corresponding text, improving their understanding of the experimental setup.

3. In the hourly tests, the model's original output was at the hourly scale, while the calibration was conducted at different time scales, and then the evaluation metrics were calculated back at the hourly scale. Why? I believe it is necessary to provide further clarification on the rationale behind this design choice.

4. In the results shown in Figure 3, the REP values are negative for most watersheds in both the daily and hourly tests. Does this mean there is a systematic underestimation of peak values in the model?

5. The paper currently concludes that higher resolution data does not necessarily improve prediction accuracy. It would strengthen the paper to include further analysis or discussion from a hydrological mechanism perspective to explore the underlying reasons for this finding.

---

## Author Response (AR1)

Dear editor and reviewers,

Thank you for your thorough review of our manuscript. We sincerely appreciate your evaluation and the constructive suggestions you provided, which certainly helped us improve the quality of our study. We have carefully considered all of your comments and revised the manuscript. The following are point-to-point response to each comment including all relevant changes made in the manuscript.

**Editor**

**Comment 1:** Three reviewers have raised several concerns and comments on how to improve this manuscript. For me, the consistency of the model's temporal scale in model output and calibration, in-depth analysis of streamflow variation with different streamflow indicators and lack of physical interpretation of the outcomes should be carefully addressed.

**Response:** Thank you for your comments and suggestions, as well as those from the three reviewers. Detailed responses to each of the reviewers' comments are provided below. Regarding your first concern about the inconsistency between the model output and calibration time scales, I would like to offer the following explanation. The purpose of this paper is to assess the value of data with different temporal resolutions in hydrological modeling. The computational time step of the model could be a factor influencing model performance, and we wanted to exclude this factor to focus on our research question. Therefore, in each test case, we set the model's computational time step uniformly to 1 hour, resulting in an original model output at an hourly scale. On the other hand, the value of observed streamflow data to hydrological models lies in its use as a benchmark for parameter calibration. This requires aggregating the model output to various time scales and comparing it with observed data at different resolutions to calibrate model parameters. Thus, when accounting for both the need to eliminate the impact of computational time step and the value of observed streamflow data at different resolutions, the inconsistency between model output and calibration time scale becomes unavoidable. We have made some revisions in Section 2.3, "Model experimental design," to clarify the reason for this inconsistency.

To address the need for more in-depth analysis and physical interpretation of the outcomes, and in line with your and the other reviewers' comments, we have selected three representative catchments and provided a detailed description in Section 3.4 of the revised manuscript. These three representative catchments were selected based on a comprehensive consideration of basin attributes including drainage area, rainfall gauging area and streamflow variability, which were identified as the most important factors impacting the model performance and its improvement. Through specific case studies, we demonstrate that the ability of increased data resolution to introduce new, accurate, and effective information is crucial in determining its potential value to enhance model performance. The GOUE index employed in this study offers a partial indication of the potential informational content that high-resolution data may contribute.

In the discussion section, in conjunction with other existing studies, we also presented several additional reasons that may limit the value of high-resolution data, including the lower signal-to-noise ratio of high-resolution data, the model's ability and suitability to capture high-resolution data, and the simplifications of real hydrological processes within the model.

**Reviewer #1**

**Comment 1:** In addition to the rainfall and runoff data already mentioned in the paper, other data used for modeling should also be detailed.

**Response:** Thank you for your reminder. Indeed, we also used DEM, potential evapotranspiration, Temperature, Normalized Difference Vegetation Index (NDVI) and Leaf Area Index (LAI) data necessary to drive the THREW model in this study. The DEM in this study was from the MERIT Digital Elevation Model (DEM) with a spatial resolution of 90m. Temperature and potential evapotranspiration data were sourced from ERA5-land. NDVI and LAI were obtained from MODIS products. These descriptions have been added to the last paragraph of Section 2.1.

**Comment 2:** It would be beneficial to reference Figure 2 at the beginning of Section 2.3 when introducing the experimental design. This will help readers refer to the flowchart while reading the corresponding text, improving their understanding of the experimental setup.

**Response:** Thanks for your suggestion. We have referred to Figure 2 at the beginning of Section 2.3 to enhance the reader's understanding of our experimental setup.

**Comment 3:** In the hourly tests, the model's original output was at the hourly scale, while the calibration was conducted at different time scales, and then the evaluation metrics were calculated back at the hourly scale. Why? I believe it is necessary to provide further clarification on the rationale behind this design choice.

**Response:** As the primary approach of this study, the experimental design indeed needs to be clearly and thoroughly explained. We apologize for not making it clear earlier. I would like to first explain the rationale behind this design:

1) Daily and hourly scales are the most concerned and commonly used scales in current hydrological modeling. The purposes of the daily and hourly tests are to investigate the effects of input data and measured streamflow with different resolutions on daily streamflow and hourly streamflow simulations, respectively. Therefore, the final evaluation metrics are the assessments of the simulated daily and hourly streamflow results.

2) To eliminate potential impacts of the model's computational time step, the time step was standardized to 1 hour across all cases in both tests. This led to the model's original output being at the hourly scale in all cases. In the hourly test, the observed streamflow was also treated as input data for model to its calibration, and its resolution was the subject of investigation. Therefore, to explore the effects of observed streamflow data at different resolutions, the model's original hourly simulated streamflow was aggregated into different time steps, which were then used for model calibration.

In Section 2.3 in the revised manuscript, when introducing the two experiments, we first clarified our research objectives, and appropriately added some detailed descriptions to ensure that readers can clearly understand the rationale behind our experimental design.

**Comment 4:** In the results shown in Figure 3, the REP values are negative for most watersheds in both the daily and hourly tests. Does this mean there is a systematic underestimation of peak values in the model?

**Response:** Accurately simulating peak flows is a challenge for all models. This is especially true for small and medium-sized basins like those in this study, where streamflow changes relatively quickly and flood durations are shorter, making it more difficult to accurately model peak flows. Results in this study also indicate that the THREW model performs better in basins with larger areas, higher precipitation, and smaller diurnal runoff variations.

In this study, peak flows were underestimated in most basins, which might be due to the use of the single-objective KGE metric during model calibration. The KGE metric reflects the model's accuracy over the entire time series. The model is a simplification of natural runoff processes and operates with fixed parameters throughout its execution. The accuracy of the model's simulation of peak flows and other processes (as evaluated by REP and KGE) are sometimes conflicting objectives. Therefore, to achieve a higher KGE, the optimization algorithm tends to select parameters that improve the overall simulation accuracy, which leads to the underestimation of peak flows in most small basins.

This study focuses on the performance trends of the model under different data resolution conditions. All experiments strictly adhered to the principle of controlling variables. We used the same model, parameter range, and parameter optimization algorithm throughout the study, and the model performed well overall. Therefore, even if there was a systematic underestimation of peak values, the trend remains consistent. Therefore, the conclusions of this study are not affected by the systematic underestimation.

**Comment 5:** The paper currently concludes that higher resolution data does not necessarily improve prediction accuracy. It would strengthen the paper to include further analysis or discussion from a hydrological mechanism perspective to explore the underlying reasons for this finding.

**Response:** Thanks for your suggestion. Other reviewers have made the same recommendations, and we recognize the need for a deeper analysis. Therefore, combined with suggestions from other reviewers, we have selected three representative catchments based on a comprehensive consideration of basin attributes including drainage area, rainfall gauging area and streamflow variability, which were identified as the most important factors impacting the model performance and its improvement. Hourly time series of rainfall, simulated flow and observed flow under 1-hour and 24-hour resolution data for the representative catchments and representative flood events have been presented in Section 3.4 of the revised manuscript.

Through the time series of observed and simulated flow at the representative catchments, this study intuitively showed that in smaller catchments with larger streamflow variations (e.g., Catchment 1), 24-hour resolution data struggles to accurately capture the true rainfall-runoff processes, and even fails to distinguish between different events. In such catchments, increasing the data resolution provides enough additional effective information, which significantly enhances the accuracy of simulations. In catchments with moderate area and diurnal streamflow variations (e.g., Catchment 2), the difference between 24-hour resolution data and observed values is relatively large, particularly for peak flows. However, this difference is markedly reduced when the data resolution is increased to 6 hours. Further increases in data resolution may not yield additional effective information, resulting in limited improvement in model accuracy. In catchments with large area and small diurnal streamflow variations (e.g., Catchment 3), the observed flow at a 24-hour resolution already closely approximates actual flow, with the relative error in peak flow being only -8.8%, compared to -62.3% and -28.9% for the peak flow errors in the other two catchments. The discrepancy is even negligible when the resolution is increased to 6-hour. Thus, when using 24-hour resolution data, the model performs well on an hourly scale, and continuous improvement in resolution did not lead to a notable increase in simulation accuracy.

In summary, from a mechanistic perspective, this study identified drainage area, rainfall gauging area and streamflow variability as key factors influencing the value of high-resolution data. Through specific case studies, we demonstrate that the ability of increased data resolution to introduce new, accurate, and effective information is crucial in determining its potential value to enhance model performance. The GOUE index employed in this study offers a partial indication of the potential informational content that high-resolution data may contribute.

In the discussion section, in conjunction with other existing studies, we also presented several additional reasons that may limit the value of high-resolution data, including the lower signal-to-noise ratio of high-resolution data, the model's ability and suitability to capture high-resolution data, and the simplifications of real hydrological processes within the model.

**Reviewer #2**

**Comment 1:** My biggest concern is this work could be better instead of just a labor work. We are wondering the differences between the simulated streamflow using higher and lower resolutions of rainfall and streamflow, e.g., the timing and the magnitude etc. At least authors should plot the streamflow time series at several representative gauges to be transparent about the results. Sometimes, the metrics cannot fully tell the true performance or even give wrong indications.

**Response:** Thank you for your constructive suggestion. We recognized the necessity of presenting representative data and conducting a more in-depth analysis. Therefore, we have selected three representative catchments based on a comprehensive consideration of catchment's attributes such as drainage area, rainfall gauge area, and streamflow variability, which were identified as the most important factors impacting the model performance and its improvement. Hourly time series of rainfall, simulated flow and observed flow under 1-hour and 24-hour resolution data for the representative catchments and representative flood events have been presented in Section 3.4 of the revised manuscript.

Through the time series of observed and simulated flow at the representative catchments, this study intuitively showed that in smaller catchments with larger streamflow variations (e.g., Catchment 1), 24-hour resolution data struggles to accurately capture the true rainfall-runoff processes, and even fails to distinguish between different events. In such catchments, increasing the data resolution provides enough additional effective information, which significantly enhances the accuracy of simulations. In catchments with moderate area and diurnal streamflow variations (e.g., Catchment 2), the difference between 24-hour resolution data and observed values is relatively large, particularly for peak flows. However, this difference is markedly reduced when the data resolution is increased to 6 hours. Further increases in data resolution may not yield additional effective information, resulting in limited improvement in model accuracy. In catchments with large area and small diurnal streamflow variations (e.g., Catchment 3), the observed flow at a 24-hour resolution already closely approximates actual flow, with the relative error in peak flow being only -8.8%, compared to -62.3% and -28.9% for the peak flow errors in the other two catchments. The discrepancy is even negligible when the resolution is increased to 6-hour. Thus, when using 24-hour resolution data, the model performs well on an hourly scale, and continuous improvement in resolution did not lead to a notable increase in simulation accuracy.

In summary, through specific case studies, it was demonstrated that the ability of increased data resolution to introduce new, accurate, and effective information is crucial in determining its potential value to enhance model performance. The GOUE index employed in this study offers a partial indication of the potential informational content that high-resolution data may contribute.

**Comment 2:** I also concern the model used and the study area, which determine the generality of the conclusions, though authors acknowledge such limitations in the manuscript.

**Response:** The generality of the conclusions to other regions and models is also a concern for us and other researchers. As we mentioned in the Discussion and Limitations sections, other authors (e.g., Ficchì et al., 2016; Reynolds et al, 2017) have reached similar conclusions in related studies, and they also mentioned concerns about generality and the need for further research. In order to verify the generality of this conclusion across different climatic zones and models, we deliberately conducted similar research in northern China using another model (this work is currently under discussion at egusphere-2024-2966), and obtained similar conclusions. In that work in northern China, we also found that for daily streamflow simulations, improvements in model performance become negligible when the resolution exceeds 12 hours. As for hourly streamflow simulations, improvements in overall flood process accuracy become negligible when the resolution of input exceeds 6 hours, while higher resolutions further enhance the precision of peak flow simulations. We have addressed this issue in the Discussion section of revised manuscript.

We also anticipate the advent of further scholarly investigations encompassing diverse geographical areas and model frameworks. We believe that as more related studies emerge, we will gain a clearer and more comprehensive understanding.

References:
Ficchì, A., Perrin, C., and Andreassian, V.: Impact of temporal resolution of inputs on hydrological model performance: An analysis based on 2400 flood events, Journal of Hydrology, 538, 454-470, 10.1016/j.jhydrol.2016.04.016, 2016.

Reynolds, J. E., Halldin, S., Xu, C. Y., Seibert, J., and Kauffeldt, A.: Sub-daily runoff predictions using parameters calibrated on the basis of data with a daily temporal resolution, Journal of Hydrology, 550, 399-411, 10.1016/j.jhydrol.2017.05.012, 2017.

Tudaji, M., Nan, Y., and Tian, F.: Assesing the Value of High-Resolution Data and Parameters Transferability Across Temporal Scales in Hydrological Modeling: A Case Study in Northern China, EGUsphere [preprint], https://doi.org/10.5194/egusphere-2024-2966, 2024.

**Comment 3:** Line23: runoff is not input.

**Response:** We apologize for the confusion caused by our wording. Our intention was to emphasize the importance of rainfall and observed streamflow data to the model. We revised the original sentence in the manuscript as follows:

"The effectiveness of these models heavily depends on the quality and resolution of the data, especially the rainfall used for forcing and measured streamflow for calibration."

**Reviewer #3**

**Comment 1:** The primary limitation of this work is the lack of physical interpretation of the outcomes. The hydrological model is treated more as a "black box," failing to provide insights into the hydrological processes that might explain the presented results. I therefore recommend this paper for major revision. In particular, I suggest enhancing the description of the hydrological model, which could assist both the authors and readers in interpreting the results in the discussion section.

**Response:** Thank you for your suggestion. We recognize that our manuscript lacked a detailed description of the model and a thorough explanation of its outputs. In Section 2.2, we have added a more comprehensive description of the model, including the parameter settings used in this study. Additionally, we selected three representative catchments and have provided a detailed discussion of the differences in simulation results under varying resolution input and streamflow data conditions. We hope this will help clarify the execution, results, and discussion of our experiments.

**Comment 2:** The methodology section is somewhat difficult to follow. From what I understand, the minimum time step used is one hour, meaning that processes are integrated over time using this interval. This implies that the maximum resolution of the input data is hourly. Once this time step is established, it is possible to produce output at this resolution or at coarser resolutions (e.g., 2 hours, 3 hours, up to 24 hours), but not finer. The discriminating factor is therefore the integration time, after which you may choose to aggregate the output (e.g., streamflow) at a daily time step to compute efficiency metrics. In this context, statements like "all input data except for rainfall were resampled to the hourly resolution" could be misleading.

**Response:** Your understanding is completely correct. We sincerely apologize for the confusion caused by our wording. We acknowledge that the use of "except for" in our original statement was incorrect; the proper term should have been "besides", and the full sentence should be "All input data besides rainfall were resampled to the hourly resolution".

The purpose of this study is to investigate the value of different data resolutions (including rainfall and streamflow) on hydrological modeling. The original resolution of the rainfall data is slightly finer than 1 hour. We first resampled the rainfall data to other temporal scales between 1 and 24 hours to generate rainfall data at different resolutions. The original resolutions of the other driving data are 24 hours. To eliminate the influence of computation time step, we fixed the model's computation time step at 1 hour. The model was run with a 1-hour computation time step, so if the input data resolution was coarser than 1 hour, it needed to be resampled to a 1-hour resolution before the calculations, and then generate results at a 1-hour resolution. Subsequently, to explore the effect of streamflow data at different resolutions, the 1-hour resolution simulation results were aggregated to other temporal scales, and the model was calibrated based on observed streamflow at the corresponding scales.

In the revised manuscript, we have made appropriate modifications to Section 2.3 on experimental design, including the use of correct wording, emphasizing the purpose, and adjusting the narrative sequence, to enhance reader comprehension of our methodology.

**Comment 3:** Please provide the equation for the GOUE. Additionally, a table outlining the influencing factors would enhance the readability of the paper.

**Response:** Thank you for your suggestion. The equation for the GOUE is shown as follows:

$$\text{GOUE} = 1 - \frac{\sum_{i=1}^{n} \left(Q_i^{Dh} - Q_i^{h}\right)^2}{\sum_{i=1}^{n} \left(Q_i^{h} - \overline{Q^h}\right)^2}$$

where n is the length of the time series of the streamflow, $Q^h$ is the actual hourly streamflow, $\overline{Q^h}$ is the mean of the $Q^h$, $Q^{Dh}$ is the hourly streamflow based on daily streamflow assuming a uniform intraday streamflow. We added the GOUE's equation and description to the revised manuscript. Besides, a brief introduction and abbreviations of the influencing factors have been included in Table 1, and we mentioned Table 1 in Section 2.1 of the revised manuscript.

**Comment 4:** Could the authors discuss other potential factors that might explain the differences in performance between the different basins? For instance, consider a scenario where a particular process is not well-modeled. This process could be highly significant in one basin but negligible in another, leading to varying results. Could this be the case for some of the basins under consideration?

**Response:** Thank you for your suggestion. As you mentioned, the accuracy of the model's simulation of a particular hydrological process and the importance of that process within the catchment could indeed be one of the potential factors. However, since the meteorological conditions across our study catchments are quite similar, there might not be a significant difference in the importance of a particular hydrological process across different catchments. Therefore, this factor may have little impact on the model's varying performance across the catchments.

**Comment 5:** The table captions need to be expanded. Specifically, the meaning of bold text and asterisks should be clearly explained in the captions.

**Response:** Thank you for your suggestion. We followed your suggestion and explained the meaning of the bold text and asterisks in the captions in the revised manuscript.

**Comment 6:** The second question regarding the time step resolution is not fully addressed. According to the methodology section, the highest resolution used is hourly. With input data of an hourly resolution, one could potentially model processes at this time step.

**Response:** We apologize for imprecise wording and not clearly explaining the focus of our study. Indeed, if high-resolution data (such as 1-hour resolution) is already available, it can typically provide reliable hourly simulations. However, the purpose of our second question and the design of our second experiment is to determine whether, in scenarios where data availability is limited, relatively coarse resolution data can still offer the same level of reliability in hourly simulations. If so, what is the coarsest resolution that maintains this reliability? We are searching for a resolution threshold: if the data resolution is coarser than this threshold, the reliability of the results decreases; if it is finer, there is no significant improvement in reliability.

We revised the second question to: "What is the coarsest resolution of rainfall and streamflow data required to provide reliable hourly streamflow simulations?" This change, replacing "necessary" with "coarsest," should help reduce confusion and enhance readability.

**Comment 7:** Line 25-28: References are required to support these statements.

**Response:** Thank you for your reminding. We have added relevant references to the revised manuscript. The following is the revised content:
To address this limitation, data is often artificially disaggregated from raw time series using mass curves (Blöschl and Sivapalan, 1995) or complex stochastic generators (Creutin and Obled, 1980). However, models based on coarsely resolved or artificially refined data can introduce biases, particularly when forecasting at finer temporal scales, as they may not accurately capture the variability and magnitude of hydrological variables (Younis et al., 2008; Huang et al., 2019).

**References:**
Blöschl G, Sivapalan M. Scale issues in hydrological modelling: a review[J]. Hydrological processes, 1995, 9(3-4): 251-290.
Creutin J D, Obled C. Modelling spatial and temporal characteristics of rainfall as input to a flood forecasting model[J]. IAHS-AISH Publication (129), 1980: 41-49.
Huang, Y., Bárdossy, A., and Zhang, K.: Sensitivity of hydrological models to temporal and spatial resolutions of rainfall data, Hydrol. Earth Syst. Sci., 23, 2647–2663, https://doi.org/10.5194/hess-23-2647-2019, 2019.
Younis J, Anquetin S, Thielen J. The benefit of high-resolution operational weather forecasts for flash flood warning[J]. Hydrology and Earth System Sciences, 2008, 12(4): 1039-1051.

---

## Author Response (AR2)

Dear editor and reviewers,

Thank you for your thorough review of our manuscript. We sincerely appreciate your comments and constructive suggestions, which helped us improve the quality of our study. We have carefully considered all of your comments and revised the manuscript. The following are point-to-point response to each comment including all relevant changes made in the manuscript.

In addition to these revisions, we have added a funding source that supported our research to the Acknowledgments section.

**Editor**

**Comment 1:** The manuscript still lacks insights into the physical processes modeled which should intrinsically linked to the specific characteristics and limitations of the model itself. Specifically, Figure 2 should be revised to clearly explain and link the different modules of the hydrological model.

**Response:** Thanks for your suggestion. Based on the original discussion, and incorporating your and the reviewers' suggestions, we have added explanations for the limited benefit of increasing data imposed by other driving data (such as evapotranspiration) and model's structure. Furthermore, we reorganized these potential causes and explanations from the perspectives of data-driven factors and model structure, providing evidence from the results of this study and other literature to better reveal how data at different resolutions influence the simulation of hydrological processes.

Figure 2 has been revised and designed as a block diagram that clearly explains and links the different modules of the hydrological model.

**Comment 2:** The authors clarify, based on the findings of this study, which of these potential reasons might be the most significant and whether there is supporting evidence within this study for these explanations.

**Response:** Based on the results of this study, among the potential causes listed, the spatial and temporal autocorrelation of the data (as discussed in the first point of the discussion section) could be the primary factor limiting further improvements in model performance. The fundamental value of high-resolution data lies in its ability to provide novel, accurate, and valuable information. The GOUE metric we employed effectively quantifies the additional information contained in actual hourly resolution data compared to data obtained by uniformly resampling daily resolution data to an hourly scale. Among various basin characteristics, GOUE showed the strongest correlation with IMP, indicating that in basins with greater streamflow variability, increased data resolution introduced more detailed and novel information, leading to greater improvements in model performance. The case studies of three representative basins also vividly demonstrate that the ability to enhance model performance critically depends on the introduction of additional effective information through higher-resolution data.

We have revised the wording in the discussion section to highlight this primary cause and its supporting evidence.

**Comment 3:** In addition, the authors should provide further justification for selecting these three catchments and the corresponding flood events, elaborating on their representativeness.

**Response:** We have revised the sections in the manuscript related to the selection of representative catchments to more clearly explain the rationale behind their selection and their representativeness. The revised content is as follows:

To better understand how rainfall and streamflow data at different resolutions specifically influence the hydrological

simulation results, we selected three representative catchments based on the Hourly Test results ($IMP_H$) and sensitive factors such as DRA, GOUE and RGA, which were identified as having a significant impact on $IMP_H$. These catchments were chosen as representative examples specifically due to their following typical characteristics and the distinct patterns they exhibit in hydrological simulation results, providing valuable insights into the influence of data resolution on model performance.

Catchment 1 (Tiantangyan): This catchment, characterized by relatively small DRA, GOUE, and RGA values, showed a significant improvement in simulation results with increased data resolution, as reflected by a large IMP.

Catchment 2 (Saitang): With medium values for DRA, GOUE, and RGA, this catchment demonstrated a gradual improvement in KGE as resolution increased, though the gains were less substantial.

Catchment 3 (Gaoan): As one of the largest catchments in terms of DRA, with relatively large GOUE and RGA, Gaoan exhibited limited improvement in performance with higher-resolution data, as indicated by a smaller IMP.

**Reviewer #1**

**Comment 1:** The manuscript still lacks insights into the physical processes modeled. Two examples include snow/glacier dynamics and evapotranspiration. Both processes exhibit significant diurnal and seasonal variations that influence streamflow, depending on factors such as basin size, the presence (or absence) of glaciers, and the distance to outlet points. These factors can affect the choice of the appropriate time step for simulations. For instance, if the model only accounts for daily evapotranspiration while the actual process varies hourly, it becomes clear that the results are limited by the model's inability to capture small temporal-scale variations in evapotranspiration accurately. A similar limitation could apply to snow processes, where finer temporal scales might be critical for realistic simulations.

**Response:** Thank you sincerely for your valuable advice and the examples you provided. In this study, we focused on the value of rainfall and streamflow data, while the resolution of other auxiliary data used in the model is fixed and relatively coarse. The resolution of other driving data could indeed be one of the factors limiting the further improvement of model performance. Our study catchments are located in southern China, where snowfall is rare, and glaciers are absent. As you pointed out, processes with significant diurnal variations, such as evapotranspiration (currently at a daily resolution), could be a factor constraining the model's ability to improve its performance at finer temporal scales.

Additionally, from a physical process's perspective, the re-infiltration process might also constrain the model's ability to achieve better performance with higher-resolution data. A real watershed acts as a low-pass filter, while hydrological models are simplified representations of actual runoff processes. Localized short-duration heavy rainfall may cause infiltration-excess runoff, but during the surface flow routing process over the hillslope, re-infiltration can occur, leading to slower runoff variations. Although the hydrological model does not explicitly account for re-infiltration, the use of coarse-resolution data effectively performs a "low-pass filtering" function, which can result in comparable or even better performance than when using high-resolution data. Similarly, other hydrological processes that have a smoothing effect on runoff but are not yet accounted for in the model, such as variations in water surface width during channel routing, could also be factors limiting the model's performance when using high-resolution data.

This study focuses on the value of high-resolution rainfall and streamflow data for semi-distributed hydrological models. The influence of above processes may require validation and quantification through a more physically-based hydrological model with higher resolution.

We have reorganized potential reasons from the perspectives of driving data and hydrological processes, and incorporated the above discussion into the revised manuscript's discussion and limitations section.

**Comment 2:** This observation is closely tied to the following suggestion regarding the figure representing the model. In other words, part of the discussion is, in my view, intrinsically linked to the specific characteristics and limitations of the model itself. I suggest revising Figure 2. The figure should be designed as a block diagram that clearly explains and links the different modules of the hydrological model. The current figure is of poor quality, with some arrows that are not even explained.

**Response:** Thanks for your suggestion. We have replaced the original Figure 2 with the following diagram to better explain the hydrological processes in the model.

[Figure]

Figure 2. Structural diagram of runoff generation processes in the THREW model

**Comment 3:** Finally, please update the captions for all tables to clarify the meaning of the * and ** symbols.

**Response:** Thank you for your suggestion. We have updated the captions for all tables to clarify the meaning of the * and ** symbols.

**Reviewer #2**

**Comment 1:** In the discussion section, the authors have presented several potential reasons and explanations for the limited impact of further increasing data resolution on the enhancement of simulation accuracy. Some of these appear to be summaries of findings from existing literature, without providing evidence from the current study. I suggest that the authors clarify, based on the findings of this study, which of these potential reasons might be the most significant and whether there is supporting evidence within this study for these explanations.

**Response:** Thank you for your suggestion. Based on the results of this study, among the potential causes listed, the spatial and temporal autocorrelation of the data (as discussed in the first point of the discussion section) could be the primary factor limiting further improvements in model performance. The fundamental value of high-resolution data lies in its ability to provide novel, accurate, and valuable information. The GOUE metric we employed effectively quantifies the additional information contained in actual hourly resolution data compared to data obtained by uniformly resampling daily resolution data to an hourly scale. Among various basin characteristics, GOUE showed the strongest correlation with IMP, indicating that in basins with greater streamflow variability, increased data resolution introduced more detailed and novel information, leading to greater improvements in model performance. The case studies of three representative basins also vividly demonstrate that the ability to enhance model performance critically depends on the introduction of additional effective information through higher-resolution data.

We have revised the wording in the discussion section to highlight this primary cause and its supporting evidence.

**Comment 2:** The paper selected three representative catchments from the total of 63 study catchments to illustrate how data of different resolutions specifically affect simulation accuracy. I recommend that the authors provide further justification for selecting these three catchments and the corresponding flood events, elaborating on their representativeness.

**Response:** Thank you for your suggestion. We have revised the sections in the manuscript related to the selection of representative catchments to more clearly explain the rationale behind their selection and their representativeness. The revised content is as follows:

To better understand how rainfall and streamflow data at different resolutions specifically influence the hydrological simulation results, we selected three representative catchments based on the Hourly Test results ($IMP_H$) and sensitive factors such as DRA, GOUE and RGA, which were identified as having a significant impact on $IMP_H$. These catchments were chosen as representative examples specifically due to their following typical characteristics and the distinct patterns they exhibit in hydrological simulation results, providing valuable insights into the influence of data resolution on model performance.

Catchment 1 (Tiantangyan): This catchment, characterized by relatively small DRA, GOUE, and RGA values, showed a significant improvement in simulation results with increased data resolution, as reflected by a large IMP.

Catchment 2 (Saitang): With medium values for DRA, GOUE, and RGA, this catchment demonstrated a gradual improvement in KGE as resolution increased, though the gains were less substantial.

Catchment 3 (Gaoan): As one of the largest catchments in terms of DRA, with relatively large GOUE and RGA, Gaoan exhibited limited improvement in performance with higher-resolution data, as indicated by a smaller IMP.